# Airborne remote sensing and in-situ measurements of atmospheric $CO_2$ to quantify point source emissions

Thomas Krings[1], Bruno Neininger[2,3], Konstantin Gerilowski[1], Sven Krautwurst[1], Michael Buchwitz[1], John P. Burrows[1], Carsten Lindemann[4], Thomas Ruhtz[4], Dirk Schüttemeyer[5], and Heinrich Bovensmann[1]

[1]Institute of Environmental Physics, University of Bremen, FB 1, P.O. Box 330440, D-28334 Bremen, Germany
[2]METAIR AG, Airfield Hausen am Albis, CH-8915 Hausen am Albis, Switzerland
[3]Zurich University of Applied Sciences, CH-8400 Winterthur, Switzerland
[4]Institute for Space Sciences, Free University of Berlin, Carl-Heinrich-Becker-Weg 6-10, D-12165 Berlin, Germany
[5]ESA / ESTEC, Keplerlaan 1, 2201 AZ Noordwijk, The Netherlands

*Correspondence to:* Thomas Krings (Thomas.Krings@iup.physik.uni-bremen.de)

**Abstract.** Reliable techniques to infer greenhouse gas emission rates from localised sources require accurate measurement and inversion approaches. In this study airborne remote sensing observations of $CO_2$ by the MAMAP instrument and airborne in-situ measurements are used to infer emission estimates of carbon dioxide released from a cluster of coal fired power plants. The study area is complex due to sources being located in close proximity and overlapping associated carbon dioxide plumes. For the analysis of in-situ data, a mass balance approach is described and applied. Whereas for the remote sensing observations an inverse Gaussian plume model is used in addition to a mass balance technique. A comparison between methods shows that results for all methods agree within 10% or better with uncertainties of 10% to 30% for cases where in-situ measurements were made for the complete vertical plume extent. The computed emissions for individual power plants are in agreement with results derived from emission factors and energy production data for the time of the overflight.

## 1 Introduction

Knowledge of emissions of the greenhouse gas carbon dioxide ($CO_2$) originating from localised sources is often inadequate (Ciais et al., 2015; NRC, 2010). Even for well monitored localised $CO_2$ emitters, there are significant differences between inventories calculated using different but plausible methods. For example, Ackerman and Sundquist (2008) found differences of more than 20% between emissions from U.S. power plants. Differences in inventories of about a factor of two were found for $CO_2$ emissions from flaring in the oil and gas production (Ciais et al., 2015).

Top down estimates of localised sources are generally obtained using airborne or ground based in-situ measurements. Recently also the use of airborne remote sensing has demonstrated the ability to accurately estimate emissions. All methods have their distinctive advantages and disadvantages. Ground based in-situ measurements are fairly low-cost. However, they do generally not sample the complete atmospheric boundary layer, which is necessary for an accurate emission estimate. Airborne in-situ allows accurate concentration and wind speed measurements from which emissions can be derived (e.g. Karion et al., 2013; Cambaliza et al., 2014; Caulton et al., 2014; Gordon et al., 2015; Lavoie et al., 2015). However, they require a dense

flight pattern, and assumptions, for example, about the layer below the lowest flight track have to be made. Furthermore, air space restrictions can interfere with required flight patterns. Airborne remote sensing of atmospheric column concentrations of $CO_2$ allows sounding of the complete boundary layer and offers the opportunity to survey large areas in short time spans (Krings et al., 2011, 2013; Thompson et al., 2015; Frankenberg et al., 2016; Thorpe et al., 2014; Tratt et al., 2014). In contrast to in-situ measurements, however, clear sky conditions are mostly required as they measure backscattered solar electromagnetic radiation, except for those instruments operating in the thermal infrared (e.g. Tratt et al., 2014). However, interpretation of thermal infrared measurements depends on the thermal contrast (e.g. Young, 2002). Wind information has to be additionally gathered for example from model data (see, for example, Krings et al., 2011) or additional in-situ wind measurements (see, for example, Krings et al., 2013).

Using the example of $CO_2$ emissions from a cluster of power plants in western Germany, top down results from airborne in-situ and remote sensing are evaluated and compared to each other, as well as to an independent bottom up estimate computed from energy production and emission factors. Additional complexity is added to the top down inverse problem since sources are partly located in close proximity to each other with overlapping $CO_2$ plumes. Generally, it is necessary to achieve a correct source attribution in the presence of multiple neighbouring sources for validation purposes.

## 2  Measurement campaign and target area

An airborne measurement campaign combining remote sensing measurements of column-averaged concentrations of $CO_2$ and $CH_4$, denoted $XCO_2$ and $XCH_4$ with in-situ measurements of atmospheric $CO_2$ and $CH_4$ concentration (mass per volume) as well as wind speed and direction in the boundary layer took place in Germany in August 2012. This campaign was carried out in the framework of ESA's Earth Explorer 8 activities for the candidate mission CarbonSat (Bovensmann et al., 2010; Buchwitz et al., 2013). The CarbonSat concept founded the basis for the $CO_2$ monitoring mission now under preparation within the European Copernicus program.

For the remote sensing of the greenhouse gases $CO_2$ and $CH_4$, MAMAP (Gerilowski et al., 2011) was flown above the boundary layer on the Cessna T207A aircraft of the Free University of Berlin. MAMAP is an airborne 2 channel NIR-SWIR grating spectrometer system for accurate measurements of gradients in column-averaged methane and carbon dioxide concentrations. It was jointly developed by the Institute of Environmental Physics / Remote Sensing (IUP/IFE), University of Bremen (Germany) and the Helmholtz Centre Potsdam, German Research Centre for Geosciences (GFZ). It was demonstrated that the instrument is able to detect and retrieve the total dry column of the greenhouse gases $CH_4$ and $CO_2$ with a precision of 0.3–0.4% ($1\sigma$) at local scales ($\approx 10\,\mathrm{km}$), and that MAMAP is an appropriate tool for detection and inversion of localised greenhouse gas emissions from aircraft (Gerilowski et al., 2011; Krings et al., 2011, 2013; Krautwurst et al., 2017).

For the in-situ airborne measurements, the small research aircraft METAIR-DIMO was flown in the boundary layer to perform measurements in the plumes emitted from the targets, to gather background concentration measurements and to perform wind measurements needed for the interpretation of the total column measurements of MAMAP. The aircraft is equipped with underwing-pods, carrying up to 50 kg scientific payload each. The standard equipment measures the meteorological parame-

ters wind (three-dimensional components in 10 Hz resolution), fast temperature and fast, redundant humidity. A two-channel aerosol counter (MetOne for >0.3 μm and >0.5 μm) can characterise the structure of the boundary layer. The chemical measurements are for $CO_2$ (redundant) and $CH_4$ corrected for $H_2O$ interference (dilution and spectroscopic; for details, see Hiller et al. (2014)) as well as CO, $O_3$, $NO_2$, $NO_x$, $NO_y$, $O_x$. The $CO_2$ is measured with three different time resolutions and accuracies,

resulting in an overall accuracy and precision of 0.4 ppm. The individual contributions are: (i) a fast (10 Hz) measurement with a short-term precision (e.g. while crossing a plume) of about 0.2 ppm, with a limited absolute accuracy of about 5 ppm, using a modified LiCor LI-7500. (ii) A more accurate continuous, but slower (0.3 Hz) reference with a modified LI-6262, with an accuracy of better than 0.5 ppm. The highest accuracy is from flask-samples, analysed at MPI Jena. This method is described in Hiller et al. (2014).

The surveying strategy was to simultaneously probe the atmospheric boundary layer with in-situ and remote sensing measurements where the MAMAP remote sensing measurements were performed via the second aircraft above the boundary layer.

While the complete campaign covered also other $CO_2$ and $CH_4$ emitting targets (Bovensmann et al., 2014) this study focuses on measurements obtained in an area with several lignite fired power plants in western Germany (see Fig. 1) on 15 August 2012. The power plants are Niederaußem, Neurath (old and new blocks) and Frimmersdorf. The remote sensing flights were

performed at about 11:50 – 13:40 UTC (that is 13:50 – 15:40 local time, CEST). The in-situ survey over the same area was conducted between about 12:15 and 14:20 UTC. Wind was blowing from South-East (145°- 148°) so that the $CO_2$ plumes of individual power plants were overlapping.

The in-situ measurements concentrate on transects at several altitude layers around the two Neurath power plants and around the extended area including Frimmersdorf power plant.

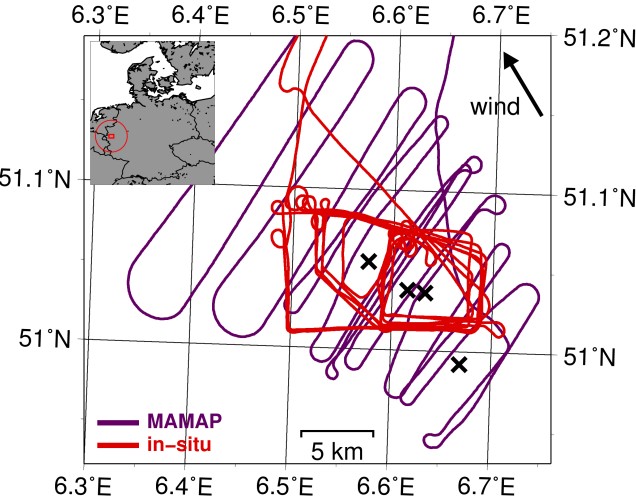

**Figure 1.** Map of the target area in western Germany. The crosses denote the four lignite fired power plants in the area. From upwind to downwind, i.e. from South-East to North-West, the power plants are Niederaußem, Neurath (new), Neurath (old) and Frimmersdorf.

## 3 Methodology

### 3.1 Flux estimates from in-situ measurements

Calculating fluxes of trace gases through an imaginary vertical plane is simple when the concentration field and the wind field are known for a sufficient time during quasi stationary conditions. However, in reality, such perfect measurements are not possible. Not the accuracy or precision of the measurements are a prime concern, but, unknown parts in the fields (inter- and extrapolations), and, most important, remaining instationarities both by short-term fluctuations (hitting a part of a plume or not), and by varying source strengths and changing meteorological conditions. Cambaliza et al. (2014), Gordon et al. (2015), Lavoie et al. (2015) and Caulton et al. (2014) are discussing comparable airborne in-situ observations downwind of regions and individual sources on different scales. The basic method to derive emissions form atmospheric $CO_2$ or $CH_4$ measurements is described in Mays et al. (2009). The instrumentation and methods that were used in the work presented here are quite similar, with the following characteristics with reference to Gordon et al. (2015): (i) A 'single screen' approach was chosen (as opposed to a box method); (ii) due to the small distance to the sources, terms like chemical conversion or storage in the volume are irrelevant; (iii) the $CO_2$ emissions were from hot stacks at short distance which reduces the problem of extrapolation to the surface; (iv) the plume concentrations were large (see Fig. 2); (v) the time resolution of the instrumentation and the relatively low speed of the aircraft (typically $40\,\mathrm{m\,s^{-1}}$) allowed to include the turbulent horizontal fluxes (at $5\,\mathrm{Hz}$ for wind, concentrations and density, the spacial resolution was better than $10\,\mathrm{m}$).

The single screen approach was chosen for practical reasons, because flying around a source means to spend most of the time in background concentrations, having a higher risk to miss quasi-stationary conditions. Some circumferential tracks (Fig. 2) confirmed the background concentrations found on the edges of the single screens. In contrast to these benefits we suffered from a non-ideal flight pattern caused by two reasons: (a) The air space restrictions both horizontally and vertically were complex, and (b) this was the first day in this yet unknown region, i.e. we were still in the process of optimizing, preparing for other days with other wind directions (south to southwest). However, this was one of only a few days of the campaign, where the remote sensing method could be directly compared with in-situ, because on many other days the cloud conditions were less ideal for remote sensing.

As in other work mentioned above, the measured data had to be inter- and extrapolated to a grid covering the whole cross sections. Because the data density on most cross sections was relatively high (see Fig. 3), and the plumes were not at all regularly shaped (rather an ensemble of puffs than a Gaussian plume), a method for the inter- and extrapolation was developed that is mainly using the grid cells of the cross sections in which measurements can be found. This structure can be seen in Fig. 4.

Before describing the interpolation process, the main difference in the method compared with other approaches is emphasised. In all the work referenced above, the fluxes were calculated by multiplying an interpolated wind field with an interpolated field of the mass distribution. This was done as well ('flux 2', as shown in Fig. 5). However, when the pointwise measurements of wind and concentration (or mass, when including density as a function of locally measured pressure, temperature and humidity) are known, the mass flux can be calculated locally, for each point of the measurements. With the mentioned common

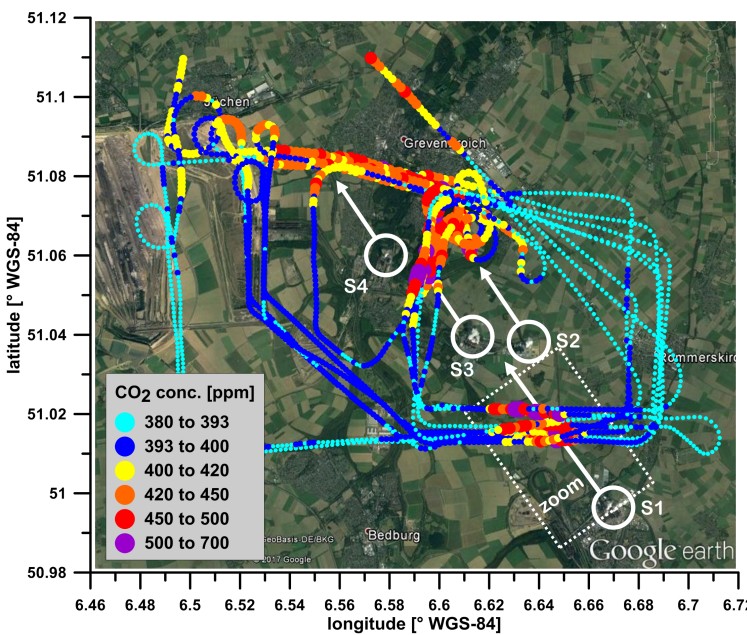

**Figure 2.** The in-situ flight pattern around and between the four sources (S1: Niederaußem; S2: Neurath (new); S3: Neurath (old); S4: Frimmersdorf), with the wind (white darts, from 146°). The $CO_2$ concentrations (1 dot per 3-s-average, i.e. about every 120 m) are color coded. In the North, the flight pattern was along an air space that could not be entered. Also on top, the cross sections were limited to 3500 ft AMSL (1050 m AMSL).

sampling rate of 5 Hz (original data 10 Hz or faster; all sensors good enough for 5 Hz), this means, that every 10 m, there is a measurement of $CO_2$ flux perpendicular to the chosen cross section, including the turbulent contribution up to this frequency or length scale. When averaging several of such measurements, this results in a direct average mass flux for a chosen grid cell in the vicinity of the flight track. The most extreme coarse approach is to average over the whole flight near the cross sec-
tion. These results are listed as 'bulk average' in the supplementary material. It is astonishing, that these results, which make no assumptions about the shape of the plumes whatsoever, and do not need any method for interpolation, are mostly near or even within the range of the more sophisticated results based on reconstructed fields. Even more extreme is 'bulk average 2', where the averages of all the mass measurements (above background) were multiplied with the averaged winds perpendicular to the whole cross sections. This finding is to some degree relativizing the discussion about subtleties in interpolation methods.
Nevertheless, in the following, we explain our linear inter- and extrapolation method, and compare it with Kriging.

    The cells shown in Figs. 4 and 7 with bold, larger numbers for the fluxes are resulting from several measurements in these cells. The inter- and extrapolation was done following three rules: (i) single gaps in columns or layers were filled by linear interpolation (first horizontal, then vertical, and again horizontal); (ii) the lowest layer with measurements was completed by taking the value of the second lowest cell above if there were valid measurements in it; (iii) larger gaps were filled differently
for two types of parameters: (a) wind and air mass (density) was filled with the averages in the layer; (b) concentrations,

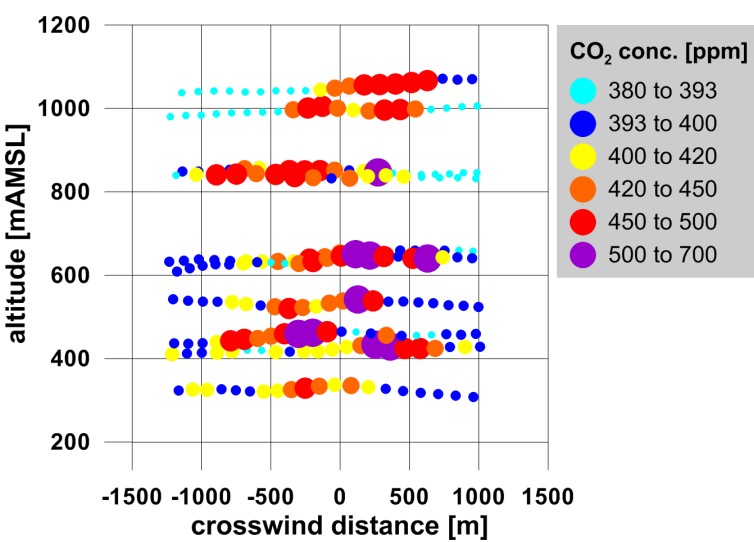

**Figure 3.** Details of the cross section through the plume of Niederaußem (S1). The projected $CO_2$ concentrations (1 dot per 3-s-average, i.e. about every 120 m) are color coded. It is important to average the fluxes along the wind, i.e. all values have to be projected to an imaginary cross section perpendicular to the wind, and not parallel to the flight pattern (see Fig. 2 and main text). The background concentration was determined to be at about 392 to 393 ppm, which is a small uncertainty compared to the maximum plume concentrations of about +100 ppm).

$CO_2$ masses and fluxes (all above background) were filled with the background. These rules are conserving the values in the cells with measurements and tend to underestimate the fluxes through cells without measurements, resulting in conservative estimates. These rules apply for cells in the range between the highest and the lowest level with measurements.

In all cases, the lowest flight track was about 150 mAGL (legal limit without special permission), and the highest track was limited by the air space, which was still in the plume on this day. This means, that the layers between 1000 mAMSL (or 1100 mAMSL for the plume Niederaußem) and the top below the stable layer at 1300 mAMSL (centered altitude of the cell), and the layers below 250 mAMSL (lower boundary of the cells at nominal 300 m) had to be extrapolated. Below the lowest layer with measurements, the fluxes were interpolated linearly to zero at the surface (using the SRTM topography for the flat terrain with an average elevation of 48 mAMSL). Since we are primarily dealing with direct flux measurements, this is the only field we have to extrapolate. However, for calculating 'flux 2' as the product of wind and mass, we extrapolated the concentrations to background at the surface and kept the wind constant. This was due to the observation, that the wind on flight altitude was usually the same as reported at the nearest airport on 10 mAGL. The areas of the cells for mass and fluxes were adapted accordingly in cells with topography. On top, the same procedure was applied: fluxes and concentrations (above background) were extrapolated to zero, with constant wind. In principal, more sophisticated extrapolation schemes could be applied as, for example, discussed in Gordon et al. (2015). Ultimately however, these profiles are not known, and their contribution here is low (less than other uncertainties).

**Niederaussem: CO$_2$ flux [kg/s] above background in the grid cells (sums and linear interpolations & extrapolations)**

| z [mAMSL] | -1200 | -1100 | -1000 | -900 | -800 | -700 | -600 | -500 | -400 | -300 | -200 | -100 | 0 | 100 | 200 | 300 | 400 | 500 | 600 | 700 | 800 | 900 | 1000 |
|---|---|---|---|---|---|---|---|---|---|---|---|---|---|---|---|---|---|---|---|---|---|---|---|
| 1300 | 0.0 | 0.0 | 0.0 | 0.0 | 0.0 | 0.0 | 0.0 | 0.0 | 0.0 | 0.0 | 0.0 | 0.0 | 0.0 | 0.0 | 0.0 | 0.0 | 0.0 | 0.0 | 0.0 | 0.0 | 0.0 | 0.0 | 0.0 |
| 1200 | 0.0 | 0.0 | 0.0 | 0.0 | 0.0 | 0.0 | 0.0 | 0.0 | 0.2 | 2.4 | 1.9 | 2.7 | 2.3 | 1.9 | 3.4 | 3.2 | 5.7 | 7.1 | 6.6 | 0.6 | 0.1 | 0.1 | 0.1 |
| 1100 | 0.0 | 0.0 | 0.0 | 0.0 | 0.0 | 0.0 | 0.0 | 0.0 | 0.5 | 4.7 | 3.8 | 5.3 | 4.6 | 3.9 | 6.8 | 6.5 | 11.4 | 14.2 | 13.1 | 1.2 | 0.3 | 0.3 | 0.1 |
| 1000 | 0.0 | 0.0 | 0.0 | 0.0 | 0.0 | 0.0 | 0.0 | 0.0 | 0.5 | 4.7 | 3.8 | 5.3 | 3.9 | 0.5 | 4.0 | 11.3 | 9.7 | 2.9 | 2.4 | 0.1 | 0.0 | 0.0 | 0.0 |
| 900 | 0.3 | 0.1 | 2.0 | 0.3 | 0.3 | 5.0 | 3.6 | 1.3 | 6.3 | 11.2 | 15.9 | 15.4 | 5.1 | 0.1 | 0.0 | 7.8 | 5.8 | 1.5 | 1.2 | 0.1 | 0.0 | 0.0 | 0.0 |
| 800 | 0.6 | 0.2 | 4.1 | 10.2 | 9.4 | 7.2 | 5.0 | 4.6 | 19.2 | 5.6 | 7.7 | 2.0 | 1.2 | 8.1 | 9.8 | 4.3 | 1.8 | 0.1 | 0.0 | 0.0 | 0.0 | 0.0 | 0.0 |
| 700 | 0.5 | 0.4 | 2.4 | 5.4 | 4.9 | 4.2 | 3.6 | 3.3 | 10.6 | 5.5 | 9.5 | 0.3 | 6.4 | 16.1 | 7.7 | 0.2 | 0.2 | 0.2 | 0.1 | 0.2 | 0.2 | 0.1 | 0.1 |
| 600 | 0.4 | 0.6 | 0.8 | 0.5 | 0.3 | 1.2 | 2.2 | 2.0 | 2.1 | 5.4 | 11.3 | 5.2 | 8.9 | 23.7 | 14.0 | 7.7 | 1.3 | 7.0 | 26.5 | 3.0 | 1.1 | 0.1 | 0.2 |
| 500 | 0.5 | 0.2 | 0.5 | 1.0 | 0.9 | 1.8 | 0.9 | 2.6 | 8.4 | 9.7 | 8.2 | 7.9 | 1.4 | 7.7 | 5.8 | 2.8 | 0.9 | 0.3 | 0.3 | 0.3 | 0.4 | 0.4 | 0.4 |
| 400 | 1.0 | 0.7 | 0.6 | 1.0 | 5.8 | 4.8 | 2.5 | 0.8 | 1.3 | 1.0 | 1.9 | 2.6 | 1.7 | 3.0 | 18.6 | 31.3 | 18.3 | 16.5 | 11.2 | 6.2 | 0.8 | 1.5 | 0.3 |
| 300 | 0.5 | 1.1 | 1.3 | 1.0 | 0.3 | 0.4 | 1.3 | 1.0 | 3.0 | 7.2 | 12.5 | 2.6 | 1.7 | 3.0 | 1.6 | 0.7 | 0.3 | 0.4 | 0.4 | 0.4 | 0.3 | 0.3 | 0.3 |
| 200 | 0.4 | 0.7 | 0.9 | 0.7 | 0.2 | 0.3 | 0.9 | 0.7 | 2.0 | 4.8 | 8.3 | 1.7 | 1.2 | 2.0 | 1.0 | 0.5 | 0.2 | 0.3 | 0.2 | 0.3 | 0.2 | 0.2 | 0.2 |
| 100 | 0.1 | 0.2 | 0.2 | 0.1 | 0.0 | 0.0 | 0.1 | 0.1 | 0.3 | 0.7 | 1.4 | 0.3 | 0.2 | 0.4 | 0.2 | 0.1 | 0.0 | 0.0 | 0.0 | 0.0 | 0.0 | 0.0 | 0.0 |
| 0 | 0.0 | 0.0 | 0.0 | 0.0 | 0.0 | 0.0 | 0.0 | 0.0 | 0.0 | 0.0 | 0.0 | 0.0 | 0.0 | 0.0 | 0.0 | 0.0 | 0.0 | 0.0 | 0.0 | 0.0 | 0.0 | 0.0 | 0.0 |
| x [m] > | -1200 | -1100 | -1000 | -900 | -800 | -700 | -600 | -500 | -400 | -300 | -200 | -100 | 0 | 100 | 200 | 300 | 400 | 500 | 600 | 700 | 800 | 900 | 1000 |

**Figure 4.** Example of results for the cells on a cross section. Shown is the direct CO$_2$ flux in the plume of Niederaußem (S1, same plume as in Fig. 3). Since the grid cells are not points, but areas filling $100 \times 100\,\mathrm{m}^2$ (or, more precise, voxels of $100 \times 100 \times 1\,\mathrm{m}^3$), the values are not displayed as contour plots, but, as pixels. The x- and z-coordinates are centered in the cells. The standard inter- and extrapolation was done with the methods described in the text. Cells with bold numbers [kg/s] are containing measurements (20 points in the average), while the inter- and extrapolated cells are shown with smaller italic numbers.

**Niederaussem: CO$_2$ flux type 2 (wind x mass) [kg/s] above background in the grid cells**

| z [mAMSL] | -1200 | -1100 | -1000 | -900 | -800 | -700 | -600 | -500 | -400 | -300 | -200 | -100 | 0 | 100 | 200 | 300 | 400 | 500 | 600 | 700 | 800 | 900 | 1000 |
|---|---|---|---|---|---|---|---|---|---|---|---|---|---|---|---|---|---|---|---|---|---|---|---|
| 1300 | 0.0 | 0.0 | 0.0 | 0.0 | 0.0 | 0.0 | 0.0 | 0.0 | 0.0 | 0.0 | 0.0 | 0.0 | 0.0 | 0.0 | 0.0 | 0.0 | 0.0 | 0.0 | 0.0 | 0.0 | 0.0 | 0.0 | 0.0 |
| 1200 | 0.0 | 0.0 | 0.0 | 0.0 | 0.0 | 0.0 | 0.0 | 0.0 | 0.2 | 2.7 | 2.4 | 3.9 | 2.3 | 2.0 | 3.4 | 3.2 | 5.6 | 7.0 | 6.7 | 0.6 | 0.1 | 0.1 | 0.1 |
| 1100 | 0.0 | 0.0 | 0.0 | 0.0 | 0.0 | 0.0 | 0.0 | 0.0 | 0.5 | 5.3 | 4.7 | 7.8 | 4.6 | 4.0 | 6.8 | 6.4 | 11.1 | 14.1 | 13.3 | 1.2 | 0.3 | 0.3 | 0.1 |
| 1000 | 0.0 | 0.0 | 0.0 | 0.0 | 0.0 | 0.0 | 0.0 | 0.0 | 0.5 | 5.3 | 4.7 | 7.8 | 2.6 | 0.6 | 4.0 | 11.3 | 9.7 | 3.0 | 2.5 | 0.1 | 0.0 | 0.0 | 0.0 |
| 900 | 0.3 | 0.1 | 1.8 | 0.3 | 0.3 | 4.9 | 3.6 | 1.3 | 6.7 | 11.2 | 16.0 | 15.3 | 4.9 | 0.1 | 0.0 | 8.0 | 6.2 | 1.7 | 1.4 | 0.1 | 0.0 | 0.0 | 0.0 |
| 800 | 0.7 | 0.2 | 3.7 | 10.2 | 9.5 | 7.2 | 4.0 | 4.5 | 19.2 | 9.6 | 7.4 | 2.0 | 2.8 | 7.8 | 8.4 | 3.6 | 2.0 | 0.1 | 0.0 | 0.0 | 0.0 | 0.0 | 0.0 |
| 700 | 0.6 | 0.5 | 2.1 | 5.0 | 5.1 | 4.6 | 3.5 | 3.5 | 10.8 | 7.2 | 10.3 | 1.1 | 6.1 | 16.4 | 7.6 | 0.2 | 0.2 | 0.2 | 0.1 | 0.2 | 0.2 | 0.1 | 0.1 |
| 600 | 0.4 | 0.6 | 0.8 | 0.5 | 0.4 | 1.2 | 2.1 | 1.9 | 2.2 | 5.2 | 12.0 | 4.9 | 8.8 | 23.7 | 13.5 | 8.3 | 1.3 | 6.6 | 26.4 | 2.9 | 1.0 | 0.1 | 0.2 |
| 500 | 0.5 | 0.2 | 0.5 | 1.0 | 0.9 | 1.8 | 1.1 | 2.9 | 8.3 | 9.0 | 6.1 | 6.7 | 2.0 | 9.4 | 6.0 | 3.3 | 1.0 | 0.3 | 0.3 | 0.3 | 0.4 | 0.4 | 0.4 |
| 400 | 1.0 | 0.7 | 0.6 | 1.0 | 5.6 | 5.1 | 2.9 | 0.8 | 1.3 | 1.0 | 2.0 | 2.6 | 1.7 | 2.8 | 18.3 | 31.4 | 20.2 | 14.3 | 11.3 | 6.2 | 0.8 | 1.5 | 0.3 |
| 300 | 0.5 | 1.1 | 1.3 | 1.0 | 0.3 | 0.4 | 1.3 | 1.0 | 2.9 | 7.4 | 13.2 | 1.6 | 4.0 | 4.8 | 1.6 | 0.7 | 0.3 | 0.4 | 0.4 | 0.4 | 0.3 | 0.3 | 0.3 |
| 200 | 0.4 | 0.7 | 0.9 | 0.7 | 0.2 | 0.3 | 0.8 | 0.7 | 2.0 | 4.9 | 8.8 | 1.1 | 2.7 | 3.2 | 1.0 | 0.5 | 0.2 | 0.3 | 0.2 | 0.3 | 0.2 | 0.2 | 0.2 |
| 100 | 0.1 | 0.2 | 0.2 | 0.1 | 0.0 | 0.0 | 0.1 | 0.1 | 0.3 | 0.8 | 1.5 | 0.2 | 0.5 | 0.6 | 0.2 | 0.1 | 0.0 | 0.0 | 0.0 | 0.0 | 0.0 | 0.0 | 0.0 |
| 0 | 0.0 | 0.0 | 0.0 | 0.0 | 0.0 | 0.0 | 0.0 | 0.0 | 0.0 | 0.0 | 0.0 | 0.0 | 0.0 | 0.0 | 0.0 | 0.0 | 0.0 | 0.0 | 0.0 | 0.0 | 0.0 | 0.0 | 0.0 |
| x [m] > | -1200 | -1100 | -1000 | -900 | -800 | -700 | -600 | -500 | -400 | -300 | -200 | -100 | 0 | 100 | 200 | 300 | 400 | 500 | 600 | 700 | 800 | 900 | 1000 |

**Figure 5.** As Fig. 4 but for CO$_2$ 'flux 2', i.e. the product of averaged wind and mass in the individual cells. The difference between these two methods is discussed in the text and is one of the sensitivity cases as summarized in Table 3, and detailed in the supplementary.

| z [mAMSL] | total CO₂ flux [kg/s] above background in the grid cells (by Kriging, without extrapolations) | | | | | | | | | | | | | | | | | | | | | | |
|---|---|---|---|---|---|---|---|---|---|---|---|---|---|---|---|---|---|---|---|---|---|---|---|
| 1300 | | | | | | | | | | | | | | | | | | | | | | | |
| 1200 | | | | | | | | | | | | | | | | | | | | | | | |
| 1100 | 0.0 | 0.0 | 0.0 | 0.0 | 0.0 | 0.0 | 0.0 | 0.0 | 0.0 | 1.5 | 1.4 | 3.0 | 3.5 | 4.0 | 5.6 | 7.9 | 10.3 | 12.8 | 9.6 | 2.5 | 0.3 | 0.2 | 0.1 |
| 1000 | 0.1 | 0.0 | 0.0 | 0.0 | 0.0 | 0.0 | 0.0 | 0.0 | 1.7 | 5.0 | 6.8 | 5.1 | 3.4 | 2.0 | 4.7 | 8.2 | 9.1 | 8.3 | 5.0 | 1.3 | 0.1 | 0.1 | 0.1 |
| 900 | 0.3 | 0.1 | 1.7 | 2.5 | 1.6 | 3.3 | 2.3 | 1.8 | 7.3 | 8.0 | 9.7 | 7.7 | 2.5 | 1.7 | 7.3 | 7.2 | 4.2 | 2.3 | 0.8 | 0.1 | 0.0 | 0.0 | 0.0 |
| 800 | 0.5 | 0.3 | 2.9 | 3.9 | 2.8 | 4.6 | 3.1 | 3.3 | 8.7 | 7.8 | 10.1 | 7.1 | 4.1 | 6.3 | 8.3 | 4.0 | 1.1 | 0.6 | 1.9 | 0.3 | 0.1 | 0.0 | 0.1 |
| 700 | 0.5 | 0.4 | 2.0 | 2.5 | 2.0 | 3.0 | 2.6 | 3.2 | 5.4 | 6.7 | 9.9 | 5.5 | 5.9 | 11.7 | 8.9 | 3.5 | 0.9 | 2.0 | 7.2 | 0.9 | 0.4 | 0.1 | 0.1 |
| 600 | 0.5 | 0.5 | 0.7 | 0.6 | 0.7 | 1.2 | 1.5 | 2.4 | 3.4 | 5.2 | 7.3 | 3.5 | 5.6 | 13.4 | 11.2 | 4.5 | 0.7 | 2.9 | 7.7 | 1.5 | 0.6 | 0.3 | 0.3 |
| 500 | 0.6 | 0.5 | 0.7 | 1.0 | 3.6 | 3.9 | 1.8 | 3.0 | 6.3 | 7.0 | 6.6 | 5.9 | 2.7 | 5.2 | 8.6 | 8.5 | 2.9 | 1.6 | 1.7 | 1.1 | 0.5 | 0.6 | 0.4 |
| 400 | 0.8 | 0.8 | 0.8 | 1.0 | 3.2 | 2.7 | 1.7 | 1.9 | 3.8 | 5.8 | 5.2 | 3.5 | 0.7 | 1.7 | 7.3 | 12.2 | 6.5 | 4.8 | 4.9 | 2.3 | 0.7 | 0.7 | 0.4 |
| 300 | 0.7 | 0.9 | 1.2 | 0.9 | 0.5 | 0.5 | 1.1 | 1.6 | 3.6 | 6.4 | 3.8 | 0.0 | 0.1 | 0.4 | 0.9 | 1.2 | 0.9 | 0.4 | 0.7 | 0.7 | 0.3 | 0.4 | 0.3 |
| 200 | | | | | | | | | | | | | | | | | | | | | | | |
| 100 | | | | | | | | | | | | | | | | | | | | | | | |
| 0 | | | | | | | | | | | | | | | | | | | | | | | |
| x [m] > | -1200 | -1100 | -1000 | -900 | -800 | -700 | -600 | -500 | -400 | -300 | -200 | -100 | 0 | 100 | 200 | 300 | 400 | 500 | 600 | 700 | 800 | 900 | 1000 |

**Figure 6.** As Fig. 4 but for the $CO_2$ flux calculated by Kriging of the original data instead of averaging in cells and applying our method for linear inter- and extrapolations. The Kriging was only done for the part of the cross section, where the layers contained data. The comparison with our standard method (see supplementary) was done by adding the percentage of the extrapolated flux (up to the stable layer and down to the surface) of the standard method (+10% in this case).

Except for the very small plume of Frimmersdorf, the contribution of the extrapolated fluxes was less than 15%. Dealing with individual local fluxes instead of mass and wind separately has an additional advantage for the extrapolation, since only one field has to be extrapolated. In cases, where the changes in wind and concentration are expected to compensate each other (e.g. increasing concentrations for sources near the surface, while the wind is decreasing), also the range of plausible values for the extrapolation has a smaller uncertainty. Whenever highly resolved data of good quality is available, we see no necessity of treating wind and concentrations (mass) separately. Neither for the cells with measurements, nor for those that need to be inter- or extrapolated.

Because the standard method for the inter- and extrapolation of measurements to a cross section in the literature referenced above is Kriging, this was applied as well (graphics program Surfer from Golden Software, including gridding with with several options for Kriging and other interpolation methods). An example of such a result is shown in Fig. 6, and all results are listed in the supplementary. By qualitative reasoning it seems, that the results are less consistent than with the rules for limited linear interpolation described above. Gordon et al. (2015) has shown that Kriging is best for simulated smooth plumes (Gaussian or similar). However, here highly irregular shapes had to be dealt with. Independent of which settings were chosen for the Kriging (the variogram for the parameter was calculated in any case and both block fits and point fits were tried) it seemed that the fields became too smooth. Kriging was applied both to the directly measured fluxes and the 'flux 2' resulting from mass- and wind-fields after Kriging. We do not claim to have found the optimum method, but, we think, that the difference between the methods is smaller than other uncertainties and that basically, we do not know which concentrations, winds and fluxes are

| z [mAMSL] | Cluster: $CO_2$ flux [kg/s] above background in the grid cells (sums and linear interpolations & extrapolations) | | | | | | | | | | | | | | | | | | | | | | | | | | |
|---|---|---|---|---|---|---|---|---|---|---|---|---|---|---|---|---|---|---|---|---|---|---|---|---|---|---|---|
| 1300 | 0.0 | 0.0 | 0.0 | 0.0 | 0.0 | 0.0 | 0.0 | 0.0 | 0.0 | 0.0 | 0.0 | 0.0 | 0.0 | 0.0 | 0.0 | 0.0 | 0.0 | 0.0 | 0.0 | 0.0 | 0.0 | 0.0 | 0.0 | 0.0 | 0.0 | 0.0 | 0.0 |
| 1200 | 0.0 | 0.0 | 0.0 | 0.0 | 0.0 | 0.0 | 2.4 | 2.0 | 3.7 | 1.1 | 1.5 | 1.1 | 2.7 | 4.9 | 4.1 | 8.3 | 5.6 | 1.8 | 1.0 | 0.1 | 6.1 | 3.3 | 2.5 | 2.7 | 1.2 | 1.0 | 0.4 |
| 1100 | 0.0 | 0.0 | 0.0 | 0.0 | 0.0 | 0.0 | 4.8 | 3.9 | 7.4 | 2.2 | 3.0 | 2.3 | 5.5 | 9.7 | 8.3 | 16.7 | 11.1 | 3.7 | 2.0 | 0.2 | 12.2 | 6.6 | 4.9 | 5.4 | 2.5 | 2.0 | 0.7 |
| 1000 | 0.0 | 0.0 | 0.1 | 0.0 | 0.0 | 0.0 | 7.2 | 5.9 | 11.2 | 3.3 | 4.4 | 3.4 | 8.2 | 14.6 | 12.4 | 25.0 | 16.7 | 5.5 | 3.0 | 0.3 | 18.3 | 9.8 | 7.4 | 8.0 | 3.7 | 3.0 | 1.1 |
| 900 | 0.0 | 0.0 | 4.1 | 8.0 | 5.9 | 12.2 | 12.2 | 8.1 | 13.3 | 9.1 | 9.6 | 8.5 | 12.4 | 17.1 | 13.1 | 18.5 | 11.7 | 8.9 | 11.8 | 12.2 | 24.7 | 18.8 | 0.0 | 0.0 | 8.0 | 3.4 | 0.0 |
| 800 | 0.0 | 0.0 | 8.2 | 16.0 | 13.3 | 9.6 | 17.3 | 10.4 | 15.4 | 14.9 | 14.8 | 13.5 | 16.5 | 19.6 | 13.7 | 12.0 | 6.7 | 12.2 | 20.6 | 24.0 | 31.2 | 2.9 | 4.0 | 5.2 | 12.4 | 3.8 | 0.0 |
| 700 | 0.0 | 0.0 | 0.0 | 11.1 | 9.8 | 14.5 | 14.0 | 10.0 | 14.2 | 16.3 | 13.0 | 11.4 | 10.9 | 18.8 | 15.9 | 6.9 | 4.7 | 24.7 | 10.4 | 12.0 | 15.6 | 1.4 | 2.1 | 2.6 | 2.9 | 3.3 | 4.1 |
| 600 | 0.0 | 0.0 | 0.0 | 6.1 | 6.3 | 19.5 | 10.8 | 9.5 | 13.0 | 17.8 | 11.1 | 9.3 | 5.2 | 17.9 | 18.2 | 1.9 | 2.8 | 1.6 | 0.2 | 0.0 | 0.0 | 0.0 | 0.2 | 0.0 | 0.0 | 0.0 | 0.0 |
| 500 | 1.1 | 0.0 | 0.0 | 0.0 | 8.9 | 13.1 | 11.0 | 2.4 | 0.6 | 3.3 | 2.6 | 3.4 | 10.2 | 8.6 | 7.0 | 4.2 | 12.5 | 16.1 | 10.0 | 9.0 | 1.8 | 0.7 | 0.2 | 2.8 | 0.0 | 0.0 | 0.0 |
| 400 | 1.0 | 1.3 | 0.0 | 0.0 | 0.5 | 0.9 | 1.0 | 1.8 | 8.3 | 14.1 | 5.3 | 6.5 | 7.6 | 9.7 | 6.1 | 10.0 | 8.8 | 1.9 | 0.7 | 1.4 | 4.7 | 2.5 | 0.2 | 0.3 | 0.1 | 0.0 | 0.0 |
| 300 | 1.0 | 1.1 | 1.1 | 0.9 | 1.2 | 3.9 | 15.2 | 1.8 | 8.3 | 14.1 | 5.3 | 4.3 | 9.9 | 3.9 | 0.8 | 0.3 | 0.3 | 0.3 | 0.3 | 0.2 | 0.2 | 0.1 | 0.3 | 0.4 | 0.1 | 0.0 | 0.0 |
| 200 | 0.7 | 0.7 | 0.7 | 0.6 | 0.8 | 2.6 | 7.0 | 1.2 | 5.6 | 9.4 | 3.5 | 2.9 | 6.6 | 2.6 | 0.6 | 0.2 | 0.2 | 0.2 | 0.2 | 0.1 | 0.1 | 0.0 | 0.2 | 0.3 | 0.0 | 0.0 | 0.0 |
| 100 | 0.2 | 0.2 | 0.2 | 0.2 | 0.3 | 0.7 | 0.0 | 0.3 | 1.9 | 3.2 | 1.2 | 1.0 | 2.2 | 0.9 | 0.2 | 0.1 | 0.1 | 0.1 | 0.1 | 0.0 | 0.0 | 0.0 | 0.1 | 0.1 | 0.0 | 0.0 | 0.0 |
| 0 | 0.0 | 0.0 | 0.0 | 0.0 | 0.0 | 0.0 | 0.0 | 0.0 | 0.0 | 0.0 | 0.0 | 0.0 | 0.0 | 0.0 | 0.0 | 0.0 | 0.0 | 0.0 | 0.0 | 0.0 | 0.0 | 0.0 | 0.0 | 0.0 | 0.0 | 0.0 | 0.0 |
| x [m] > | -2800 | -2600 | -2400 | -2200 | -2000 | -1800 | -1600 | -1400 | -1200 | -1000 | -800 | -600 | -400 | -200 | 0 | 200 | 400 | 600 | 800 | 1000 | 1200 | 1400 | 1600 | 1800 | 2000 | 2200 | 2400 |

↑
Frimmersdorf
1.23 Mt/a
39.10    kg/s

**Figure 7.** The same as Fig. 4, however for the mixed plume of the whole cluster on the northern transect. The weak plume from Frimmersdorf, which is only 4 km away from this cross section could be isolated below 450 mAMSL. The confirmation was done with an analysis with higher resolution. The layer between 350 and 450 m (centered at 400 m) has enough measurement points allowing to separate the small lower plume from the main plume from Niederaußem (probably the whole width of about 4.5 km) and the two plumes from Neurath old and new. See Fig. 8 for the separation on a cross section upwind of Frimmersdorf, closer to the three other plumes.

present between the cells with measurements. However, with the rules described above, we make sure that the values in the cells with enough measurements are not affected.

Referring to the importance of turbulent fluxes, Foken et al. (2009) gives a concise overview about the difficulties of complete closures of fluxes. The method with the direct local calculation of fluxes at each measuring point is including the turbulent fluxes in the mean wind direction. In some convective situations, also the vertical turbulent flux above the source can be considered, which was not a necessity in this case. In principle also the cross-wind turbulent diffusion could be calculated. However, this does not contribute to the flux from the source, and would only deliver an estimate for the plume broadening (lateral entrainment and detrainment). The results in the detailed table in the supplementary (difference between '200 × 100 fb' and 'flux 2') are showing, that the contribution of the turbulent fluxes is not positive, and in all the cases only a few percent. Two reasons are possible in combination: (i) uncertainties in the calculations; (ii) when the turbulent flux is mainly responsible for dilution (entrainment), the turbulence is reducing the net flux.

The positions of the cross-sections were selected (filtered from the whole flight) with minimum and maximum distance to the source, the mean wind direction, and the lateral distance from the centerline. The angle of the cross-section was adjusted for a cross-wind component of less than $0.1 \, \mathrm{m \, s^{-1}}$, and the width of the cross section should include enough background concentrations.

The background concentration for this case study was relatively easy to find, and the results are not sensitive to it because the peak plume concentrations were high above the background (100 ppm and more; see Fig. 2). Originally, the standard method was to find the background on each layer by finding the minimum concentration. However, a constant background concentration of 392 ppm was selected in order to avoid artefacts in widely contaminated layers. There is another reason against taking the

minimum for each layer, resulting typically in slightly decreasing background concentrations with altitude in this region: The emissions were injected at low altitudes into the background concentrations that were present there. When such a plume is rising into lower (or higher) background concentrations, taking the local background would lead to an over- or underestimation of the emission. Therefore it is better to take the background concentration on the altitude of the sources for the whole cross section.

Fig. 2 can also be used to explain the steps of the processing. Based on the flight track on the map, the minimum and maximum distance from the source was defined. Within these distances, the measurements are projected onto a vertical plane perpendicular to the mean wind vector. The fluxes from Niederaußem were calculated using measurements between 2 and 5 km distance, on a cross section in 3.5 km distance (this distance is not relevant, because the projection is parallel). The orientation of the cross section was adjusted until the amount of the average crosswind component was less than $0.1 \, \mathrm{m \, s^{-1}}$. Larger crosswind

components would cause artefacts by the same reason as would be the case when the cross section would be aligned with the flight track, applying wind components perpendicular to this plane. If all the flight tracks would perfectly overlap, the orientation of the cross section would not be important, and both options would be possible (exactly perpendicular to the wind, or along the flight track). However, when different tracks in different distances are involved, the maximum concentrations on the different traverses would not be aligned and would not be averaged in the same grid cell of the cross section. Instead,

they would contribute to neighboring cells, increasing these fluxes. Since the flight track in other sectors was quite complex, observing this rule was very important. Another precaution was applied: Even if the wind measurement should be accurate also in steep turns, parts with more than five degrees bank angle (roll) were excluded. Since most parts of the plumes were crossed on straight flight legs, this did not reduce the available data considerably.

For the separation of the individual contributions of sources that were emitting in the same cross section, two methods were

applied: In the case of Frimmersdorf, the small plume was identified on the cross section in the lee of all the sources. The parameters height and width of the cross section could then be adjusted to cover this plume only. This part on the larger cross section is marked in Fig. 7. However, on the cross section 'cluster 3' (see Fig. 8), the three plumes (Niederaußem, Neurath new and old) were overlapping in a way that a direct separation was not possible. Therefore, two regions on the cross section were defined, where most likely the sources Neurath old and new were dominating. The source Niederaußem, 9 km upwind of this

cross-section, was most likely contaminating this whole cross-section, mainly above 450 m, laterally within the limits marked with dashed lines. The percentage of contributions in the overlapping parts was varied in a wide range between 20 and 80% (from Neurath old or new vs. Niederaußem).

Using the difference between the whole cluster (measured directly on the most downwind cross-section), and 'cluster 3' (all power plants except Frimmersdorf) measured directly plus the small plume of Frimmersdorf it was possible to estimate the flux

that was not captured on the cross-section 'cluster 3'. The underestimation is due to a rather cautious extrapolation above the

| z [mAMSL] | Cluster 3: CO₂ flux [kg/s] above background in the grid cells (sums and linear interpolations & extrapolations) |

Table — Cluster 3: $CO_2$ flux [kg/s] above background in the grid cells (sums and linear interpolations & extrapolations)

| z [mAMSL] \ x [m] | -2200 | -2000 | -1800 | -1600 | -1400 | -1200 | -1000 | -800 | -600 | -400 | -200 | 0 | 200 | 400 | 600 | 800 | 1000 | 1200 | 1400 | 1600 | 1800 | 2000 | 2200 |
|---|---|---|---|---|---|---|---|---|---|---|---|---|---|---|---|---|---|---|---|---|---|---|---|
| 1300 | 0 | 0 | 0 | 0 | 0 | 0 | 0 | 0 | 0 | 0 | 0 | 0 | 0 | 0 | 0 | 0 | 0 | 0 | 0 | 0 | 0 | 0 | 0 |
| 1200 | 0 | 0 | 0 | 0 | 0 | 0 | 1 | 2 | 0 | 5 | 5 | 2 | 1 | 7 | 4 | 4 | 2 | 2 | 1 | 1 | 1 | 2 | 0 |
| 1100 | 0 | 0 | 0 | 0 | 0 | 1 | 2 | 4 | 1 | 10 | 10 | 3 | 2 | 14 | 8 | 8 | 5 | 3 | 2 | 2 | 2 | 4 | 0 |
| 1000 | 0 | 0 | 0 | 0 | 0 | 1 | 4 | 6 | 1 | 16 | 14 | 5 | 3 | 21 | 11 | 11 | 7 | 5 | 3 | 3 | 2 | 5 | 0 |
| 900 | 0 | 0 | 0 | 0 | 0 | 1 | 2 | 5 | 1 | 12 | 0 | 0 | 0 | 29 | 32 | 1 | 0 | 2 | 1 | 0 | 0 | 0 | 0 |
| 800 | 0 | 0 | 0 | 0 | 0 | 0 | 0 | 5 | 0 | 8 | 0 | 0 | 0 | 38 | 10 | 8 | 0 | 0 | 3 | 11 | 5 | 0 | 0 |
| 700 | 0 | 3 | 1 | 0 | 0 | 1 | 17 | 13 | 11 | 9 | 16 | 23 | 21 | 19 | 5 | 4 | 0 | 0 | 0 | 0 | 0 | 2 | 0 |
| 600 | 1 | 7 | 2 | 0 | 0 | 3 | 33 | 20 | 22 | 9 | 11 | 9 | 4 | 0 | 0 | 0 | 0 | 0 | 0 | 2 | 0 | 0 | 0 |
| 500 | 0 | 1 | 1 | 1 | 2 | 12 | 30 | 17 | 13 | 7 | 6 | 4 | 3 | 1 | 1 | 0 | 3 | 0 | 0 | 0 | 0 | 1 | 0 |
| 400 | 0 | 2 | 4 | 12 | 35 | 37 | 27 | 13 | 4 | 4 | 1 | 0 | 1 | 3 | 11 | 5 | 0 | 0 | 0 | 1 | 0 | 0 | 0 |
| 300 | 0 | 2 | 4 | 12 | 35 | 37 | 27 | 13 | 4 | 4 | 1 | 0 | 0 | 0 | 0 | 0 | 0 | 1 | 1 | 0 | 0 | 0 | 0 |
| 200 | 0 | 1 | 3 | 8 | 23 | 24 | 18 | 9 | 3 | 3 | 1 | 0 | 0 | 0 | 0 | 0 | 0 | 0 | 0 | 0 | 0 | 0 | 0 |
| 100 | 0 | 0 | 1 | 3 | 9 | 9 | 7 | 3 | 1 | 1 | 0 | 0 | 0 | 0 | 0 | 0 | 0 | 0 | 0 | 0 | 0 | 0 | 0 |
| 0 | 0 | 0 | 0 | 0 | 0 | 0 | 0 | 0 | 0 | 0 | 0 | 0 | 0 | 0 | 0 | 0 | 0 | 0 | 0 | 0 | 0 | 0 | 0 |

| overlap | Neurath old | total | Neurath new | overlap |
|---|---|---|---|---|
| 6.9 Mt/a | 20.3 Mt/a | 38.7 Mt/a | 12.3 Mt/a | 9.9 Mt/a |
| 217.5 kg/s | 642.3 kg/s | 1226.6 kg/s | 388.9 kg/s | 312.8 kg/s |

**These sums are before correcting with overlap and missing flux in the whole cross section!**

**Figure 8.** The same as Figs. 4 and 7, on the cross section downwind of the total cluster without Frimmersdorf (S4). The two domains and the overlap with the plume from Niederaußem (S1) in the background was used to separate the almost collinear plumes of Niederaußem and Neurath old (S2) and new (S3). The uncertainty was quantified by varying the parameters in the overlapping cells between 20 and 80% (background from Niederaußem versus Neurath old and new).

highest flight track. Of course also the direct measurement of the whole cluster could be underestimated by the same reason, but by adjusting the two, the budget is coherent, as a lower limit of the real emissions. Of course all these operations are not exact. Therefore the error estimates were done quite conservatively by taking the minima and maxima of the results of all the assumptions (e.g. the percentage mentioned above), resulting in error estimates of 22% for Neurath old and 50% for Neurath new. The difference is due to the fact that Neurath old is closer to the cross-section flown than Neurath new, and was therefore better distinguishable on the cross-section.

## 3.2 Discussion about main uncertainties in-situ

### 3.2.1 Measurement errors

The wind components $(u, v, w)$ have an accuracy of $0.5\,\mathrm{m\,s^{-1}}$ each, while the $CO_2$ concentrations have an accuracy of better than 1 ppm. However, as the background concentrations are subtracted, the absolute concentrations are not important, resulting in uncertainties in terms of precision (stability of the sensitivity within an hour) instead of absolute accuracies. This is resulting in maximum uncertainties of 0.5 ppm for $CO_2$ (using the two $CO_2$ instruments in combination with flask samples). The uncertainties of the wind measurements remain in the order of $0.5\,\mathrm{m\,s^{-1}}$ per component (as for the perpendicular component on the cross section). The main uncertainty is the crosswind component relative to the aircraft. However, when flying back and forth through any plume on a similar altitude, then this error is averaged out. Since this was not always possible, we are taking the worst case $0.5\,\mathrm{m\,s^{-1}}$. Then the relative error based on the wind measurement is 10% at wind speeds in the order of $5\,\mathrm{m\,s^{-1}}$, increasing with weaker winds, and decreasing with higher wind speeds. For $CO_2$, a plume with moderate 50 ppm above background only adds another percent. In the case discussed here, wind speeds were around $8\,\mathrm{m\,s^{-1}}$, with excessive $CO_2$ concentrations of more than 100 ppm (see Figs. 2 and 3), leading to the conclusion that the error for horizontal fluxes due to the uncertainties of the primary measurements is clearly below 10%. This is in agreement with the assessment found in Gordon et al. (2015). In summary: The measurement itself - if state-of-the-art - is not the main source of errors.

### 3.2.2 Other sources for errors

The accuracy and precision discussed above are important, however, not the main criterion for reliable flux estimates. Only under unrealistically ideal conditions, where winds and concentrations over the whole cross sections could be captured completely and instantaneously, and could be repeated many times, the remaining uncertainties would arise from the (systematic) measurement errors. These are, as discussed above, quite small. However, as already mentioned in Sect. 3.1, and discussed in similar work referenced there, other reasons for uncertainties can be dominating, which are harder to quantify. The approach used here was to vary the assumptions and parameters used in the calculations in a wide range. This sensitivity analysis allowed comparing the methodological variations with the average results.

The main uncertainties are dependent on the meteorological conditions and the flight patterns. An ideal flight pattern is covering the plume as completely as possible. For the $CO_2$ plumes, the minimum height of 50 m or 150 m above ground is generally sufficient when measuring close to the elevated hot sources. The main source for uncertainties in the flux estimates are the interpolations and extrapolations as discussed in Sect. 3.1.

Another source of uncertainty already mentioned is instationarity, i.e. varying source strengths from day to day, or even by the hour, while the measurements account only for the emissions for the time of the overflight. For this case study, the source variation based on energy production was less than 0.5% for Niederaußem, Neurath new and old blocks. For Frimmersdorf the variability was about 4% but with considerably lower total fluxes (see Sect. 5). Another type of instationarity is caused by the atmospheric turbulence on the scale of a few hundred meters, where a maximum concentration (puff instead of continuous plume) can be missed (=underestimation), or captured by coincidence (=overestimation). The only way to minimize this is

repeating the pattern as often as possible, and is another reason to spend as much time as possible on the single screens (cross sections). Generally spoken, the 4-d inhomogeneities cannot be captured in a "snapshot".

Another issue is the definition of the boundary layer. In an ideal case, cross sections are flown to an altitude, where no concentrations in excess of the background are detected anymore, i.e. where the plume is confined below. Such a plume was captured under ideal conditions with no extrapolation needed at all on August 23 near a single source. Therefore, this source "Weisweiler" is listed in the results Table 3 in Sect. 4.1 as a reference. The astonishing conclusion is, that the variation due to different interpolations is comparable to the non-ideal cases in this study, where extrapolation was necessary. However, the uncertainty of the extrapolations has to be added, as it is done in Table 3.

Since a deductive error estimate as it is possible for the basic measurements is not possible for the overall flux results, a sensitivity analysis was applied to all cases. In this case study, the five individual cross sections with fluxes were calculated using six sets of parameters for our approach, and for Kriging, and with three extreme (unrealistic) assumptions. The percentage of the contributions from the extrapolations is listed in Table 3 as well. They ranged between 10 and 20% for the directly measured plumes, with one exception (Frimmersdorf), where the weak plume increased the extrapolated contribution to 45%. A conservative approach for the total error is to add half of these contributions (assuming a 50% uncertainty in the extrapolations) to the differences found by the sensitivity analysis. The overall uncertainty for the total emissions of the cluster would then be 18%, 14% for Niederaußem, while ranging between 32 and 63% for other individual sources (Table 3), which were separated indirectly.

When combining sources by adding or subtracting (e.g. subtracting Frimmersdorf and Niederaußem in order to get an estimate for old and new Neurath), these uncertainties are adding in an unknown way (some systematic errors do not add, but compensate, others are adding as components (root of sums of squares), or have to be added for a possible maximum error. Therefore, when separating sources as described in Sect. 3.1, the extreme values within the parameter space were taken.

### 3.3 Fluxes from remote sensing greenhouse gas information

### 3.3.1 Measurement data

The processing of the MAMAP remote sensing data is based on the methods described by Krings et al. (2011, 2013). A modified version of the Weighting Function Modified Differential Optical Absorption Spectroscopy (WFM-DOAS) algorithm (Buchwitz et al., 2000) is used to obtain vertical column information of $CH_4$ or $CO_2$. It relies on a least squares fit of the logarithmic simulated radiance spectrum to the measurements after correction for dark signal and pixel-to-pixel gain. Additionally a look-up-table approach has been implemented accounting for varying solar zenith angle (SZA) and surface elevation. The conversion from total columns to column averaged dry air mole fractions ($XCH_4$, $XCO_2$) is performed using the proxy method, assuming that locally $CH_4$ is sufficiently constant to compute $XCO_2$ (or vice versa for $XCH_4$). See Frankenberg et al. (2005) and Schepers et al. (2012) for more information on the proxy method and Krings et al. (2011) for its application on MAMAP measurements. This method is suitable for point sources as is the case in this study.

The corresponding background profiles for the linearisation points use the U.S. standard atmosphere (U.S. Committee on Extension to the Standard Atmosphere, 1976) scaled to actual values. In this case, background $XCO_2$ was determined to 390 ppm based on in-situ data upwind of the power plants in the boundary layer and results from the SECM model (Reuter et al., 2012) above. The methane background $XCH_4$ was estimated to about 1.805 ppm also based on in-situ measurements in the upwind area of the boundary layer scaling the standard atmosphere. An assumed uncertainty of the ratio of the background columns of 1% accounts for possible deviations from these values. The spectroscopic data base used for the computation of absorption cross sections was HITRAN 2012 (Rothman et al., 2013).

Aircraft altitude during the measurements was almost constant at about 1590 mAMSL (+/- 25 m), which was also selected for the reference radiative transfer. Assuming a constant aircraft velocity of 200 km/h, the ground scene size is about 22 m x 54 m (cross track × along track) for the installed optical front telescope. Thereby the along track ground scene size describes the full width at half maximum for the sensitivity along the flight track. During the measurement, the solar zenith angle varied from about 37.5° to 45.3°.

The radiative transfer model was interpolated using a two dimensional look-up table (LUT) based on solar zenith angle and surface elevation. For that the SRTM digital elevation model (Shuttle Radar Topography Mission (SRTM) version 2.1, http://dds.cr.usgs.gov/srtm/version2_1/, a collaborative effort from NASA, NGA as well as the German and Italian Space Agencies) was used. Due to the changing measurement geometry, the conversion factor to correct for the altitude sensitivity effect (Krings et al., 2011) has to be determined for each measurement independently using also a LUT. This correction takes into account that light passes twice below the aircraft where the observed plumes are located. On average, the conversion factor for the present measurements is about 0.49.

### 3.3.2 Quality filtering

Filtering, based on the spectroscopic fit quality, has been applied rejecting measurements with a root mean square (RMS) value of the differences between measurement and model after the fit larger than 0.9% relative to the model affecting about 0.1% of the total measurements. The threshold was empirically determined from the distribution of RMS values ordered by size (compare also Krings et al., 2011, 2013).

An additional filter has been applied dependent on the signal strength to avoid measurements close to saturation (more than ≈90% detector filling) and in the lower signal to noise range, e.g., over water which has a lower surface spectral reflectance in the short-wave infrared. Filtering of the data accounts for not only SNR but also whether linear full well is achieved. For the full well ADC range selected by the manufacturer a non-linear behavior could be observed for very high detector fillings. Therefore data with very high filling factors are excluded from further processing. However, out of all measurements, the selected maximum threshold value affects only 4 single measurements during the whole measurement period.

Measurements with a detector filling of less than about 20% (13000 counts) appear to have a slight signal dependency and were neglected for the inversion process. Furthermore to ensure nadir viewing geometry the deviation from the vertical was not allowed to exceed 5°.

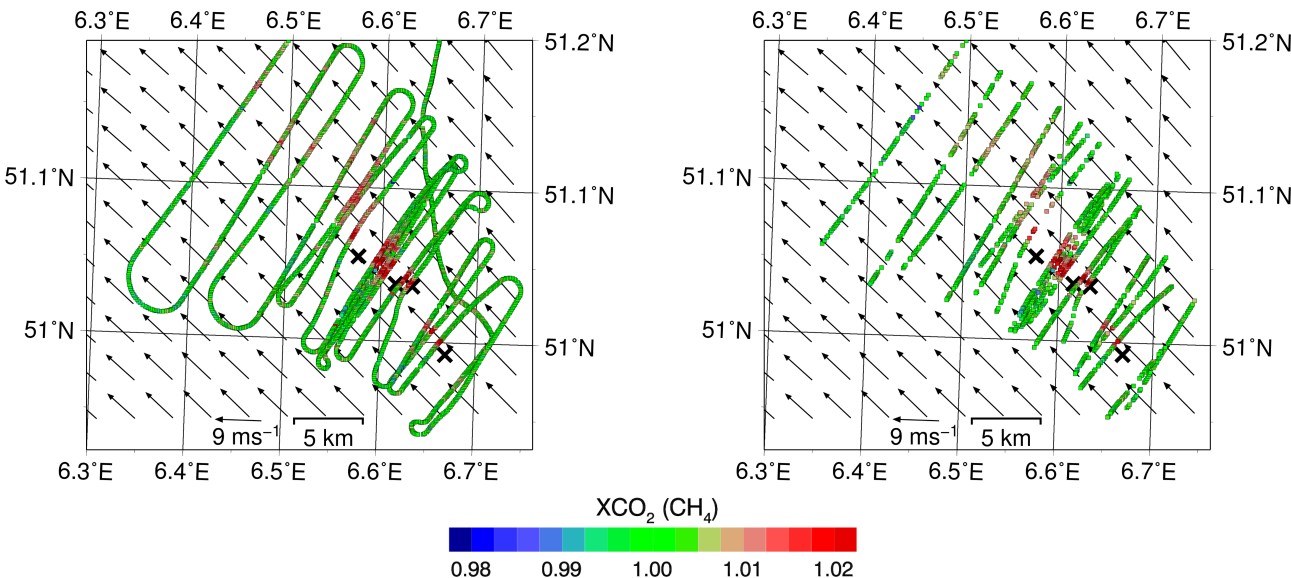

**Figure 9.** Qualitative MAMAP remote sensing $XCO_2$ data unfiltered (left) and filtered as described in the main text (right). The crosses denote the power plant locations (see Fig. 1). Arrows denote wind vectors from the COSMO-DE model at an altitude of about 350 mAMSL (model layer 45) at 13:00 UTC.

The $XCO_2(CH_4)$ precision after filtering is approximately 0.29% determined from the standard deviation of the data outside the plume area.

Fig. 9 shows the $XCO_2(CH_4)$ data acquired over the coal fired power plants without and with the filtering applied. Clearly visible are the overlapping $CO_2$ plumes originating at the individual power plant locations and advected in downwind direc-
tion towards North-West in agreement with the wind field as computed by the the routine analysis of the numerical weather prediction model COSMO-DE (Doms, 2011).

### 3.3.3  Atmospheric conditions and wind information

A fundamental parameter for the inversion is wind speed. To compute an average wind speed throughout the $CO_2$ plume from model and in-situ data also the boundary layer depth is important.
The aerosol and virtual potential temperature profiles give no indication that the transition to the free troposphere is located in the lower 1100 m. This is furthermore confirmed by the fact that there are enhanced $CO_2$ amounts throughout the probed altitude layers. Consequently it can only be concluded that the boundary layer depth is likely larger than 1100 m. For the analysis a boundary layer depth of 1500 m was assumed (with uncertainty estimates for cases of 1200 m and 1800 m, see Sect. 3.4).
Figure 10 shows measured wind speed and direction for the day analysed in this study compared to results from the COSMO-DE model. The measured in-situ wind speed averaged over 60 s ranges from about $7\,\mathrm{m\,s^{-1}}$ to $11\,\mathrm{m\,s^{-1}}$ for the time and altitude

**Figure 10.** Wind speed (left) and direction (right) as measured from the in-situ turbulence probe as well as COSMO-DE model data and surface in-situ information from Düsseldorf airport about 30 km north-north-east of Neurath power plant (obtained from weather underground, http://www.wunderground.com).

range of the overflight and is rather constant with altitude. Altitudes covered by in-situ measurements range from about 300 m to 1100 mAMSL.

Modelled and measured wind direction are similar in lower altitudes but agree less well at higher altitudes and later times (Fig. 10). However, the wind direction is derived from the remote sensing data directly without using the model information.

**Table 1.** Comparison of modeled and measured wind speed for several altitude layers.

| Altitude range | Model wind speed | In-situ wind speed | Wind speed difference (in-situ - model) | Relative wind speed difference (in-situ - model)/model |
|---|---|---|---|---|
| [mAMSL] | $[\mathrm{m\,s^{-1}}]$ | $[\mathrm{m\,s^{-1}}]$ | $[\mathrm{m\,s^{-1}}]$ | [%] |
| 291 – 440 | 8.99 | 8.53 | -0.46 | -5.1 |
| 440 – 588 | 9.33 | 8.47 | -0.86 | -9.2 |
| 588 – 737 | 9.48 | 8.53 | -0.95 | -10.0 |
| 737 – 885 | 9.84 | 9.11 | -0.74 | -7.5 |
| 885 – 1034 | 9.27 | 9.40 | +0.13 | +1.4 |
| | | | Average: | Average: |
| | | | $-0.58\mathrm{m\,s^{-1}}$ | -6.1% |

Since in-situ wind information is not always available in time and space where remote sensing measurements are taken, the in-situ wind data is used to calibrate the COSMO-DE model result in the measurement area during the time of the remote sensing overflights.

The precision of the wind model was estimated to about $0.9\,\mathrm{m\,s^{-1}}$ ($1\,\sigma$) with negligible bias for the case described in Krings et al. (2011). Assuming the same error holds in the present study, this leads to a wind based relative error ($1\,\sigma$) on the inversion of about 10%. This error can be reduced when on site wind information is available, for example, from airborne turbulence measurements as they were used in Krings et al. (2013) and that were also performed during the present campaign.

To quantify the difference between measured and modeled wind speed, the probed altitude range has been divided into 5 equally thick layers in which the deviation between the in-situ measurements on the one hand and the associated model data interpolated in time and space on the other hand were computed and subsequently averaged over the altitude layers.

For the available model and measurement data from the target area and time, this yields an average overestimation of about $0.58\,\mathrm{m\,s^{-1}}$, i.e. about 6.1%. This is well within the approximate error of about $0.9\,\mathrm{m\,s^{-1}}$. Similar to Krings et al. (2013) the in-situ wind error of $0.5\,\mathrm{m\,s^{-1}}$ is assigned to the calibrated wind.

The complete results are shown in Table 1. Note that the results do not directly relate to Fig. 10 which only shows the model wind speed at one specific COSMO-DE grid point while model wind data from the whole measurement area enters the computations in Table 1. The standard deviation of the model wind speed over the measurement area for the model layer shown in Fig. 10 is about 5.8%.

### 3.3.4 Inversion for emission data

Emission rate estimates are then obtained using an inverse Gaussian plume model fitting flux and atmospheric stability. In a second approach mass balance estimates are computed leading to two independent inversion methodologies with the exception of wind information which is taken from COSMO-DE model and the in-situ turbulence probe of the DIMO aircraft and that is used for both methods.

Preparation and performance of the Gaussian plume inversion and the integral inversion method is very much in line with Krings et al (2011, 2013). Since the inversion proved to be extremely stable no a priori information on emission rate or stability were required simplifying the cost function to be minimised in the iterative inversion process.

The data were gridded to pixels of $35\,\mathrm{m} \times 35\,\mathrm{m}$ having approximately the same area as the MAMAP ground scene size. The impact of different pixel sizes for the gridding is assessed in Sect. 3.4. No additional smoothing was applied. Note that the gridding was not used for the mass balance method.

Prior to the inversion the data were normalised dividing by the regional background. Since the measured area and time is somewhat larger than in the previous studies, no constant normalisation was selected but a track by track procedure that is also able to account, for example, for linear gradients that are unrelated to the source. Thereby data from each cross section (see Fig. 9) is normalised by a linear function that is determined by the flanks of the track excluding the plume area. If the track does not measure sufficient data outside the plume, then this method results in an underestimation of emissions. Because of that and because gaps due to filtering are rather large from track 10 onwards, only the first 9 downwind tracks with respect

to Niederaußem power plant could be analysed using the standard filtering. To investigate the measurements from further downwind, the signal threshold was relaxed for the first three tracks downwind of power plant Frimmersdorf to a minimum of 3000 counts and the inclination filter to 15°. In this way it is ensured that a sufficient set of measurements, even if of lower quality, are available for interpretation. The Gaussian plume method was not applied for these data as that would require to mix data which were subject to different filter criteria.

Obtaining an adequate estimate of the mean wind speed with which the emitted gas is transported is generally challenging when there are several sources which are separated in downwind direction. In the present case study the low variability of wind speed with altitude (see Fig. 10), however, makes the estimation less sensitive to errors in this regard. Here, the mean wind speed was estimated assuming Niederaußem power plant, the power plant that is located most upwind of all emitters, as an only source. The emitted $CO_2$ was then distributed using a vertical Gaussian dispersion with the stability parameter resulting from 2D horizontal Gaussian plume inversion model. This information could be used to obtain an altitude weighted mean wind speed for the remote sensing flight legs based on relative concentrations per altitude layer. The resulting wind speeds for each remote sensing flight leg were used individually for the mass balance approach and averaged for the inverse plume model over the relevant area. At the first remote sensing leg 700 m downwind of power plant Niederaußem, the plume reaches about 1 km height, and at 2 km downwind distance the $CO_2$ is already well mixed according to the plume model which represents a temporal average.

Boundary layer top height (represented as a reflective layer as in Turner, 1970) and emission height needed for the vertical dispersion were varied around the baseline parameters to estimate the range of errors resulting from these assumptions (see Sect. 3.4).

## 3.4 Error assessment

The influence of various parameters on the inversion results was investigated. This was mainly accomplished by evaluating errors introduced by uncertainties in the input parameters for the inversion methods, except for the statistical error on the plume inversion which yields about 3% for the total emission estimate and about 6% maximum error for a single power plant emission and for the statistical error on the mass balance approach derived from variability of inversion results (see Sect. 4.2).

The wind direction of 147.5° for the inversion was determined by visually matching the best wind direction and minimizing the retrieved stability parameter $a$ of the Gaussian plume inversion and hence minimizing the plume width relative to the centre axis (see Fig. 11). This also minimizes the total emission rate. However this minimum does not exactly coincide with the best fit (lowest RMS of differences between model and measurements) which is accomplished for about 145.5°. To account for the fact that the minimum in stability and fit quality is rather wide, an error of about ±4° was assumed. Importantly, Fig. 11 shows furthermore that, in principal, wind direction can be fitted directly to the data.

Several quality filters were applied to the data. The filter for fit quality (see Sect. 3.3.2) has been set relatively broad to reject only the data of poorest quality. It does not improve the results to relax the filter further or apply a stricter criteria to reject data. The signal filter was set to reject data with a signal below 13000 counts (see Sect. 3.3.2). When relaxing the filter criteria by −2000 counts the total emission estimate is affected by about 6% for the plume inversion and about 1% for the mass balance

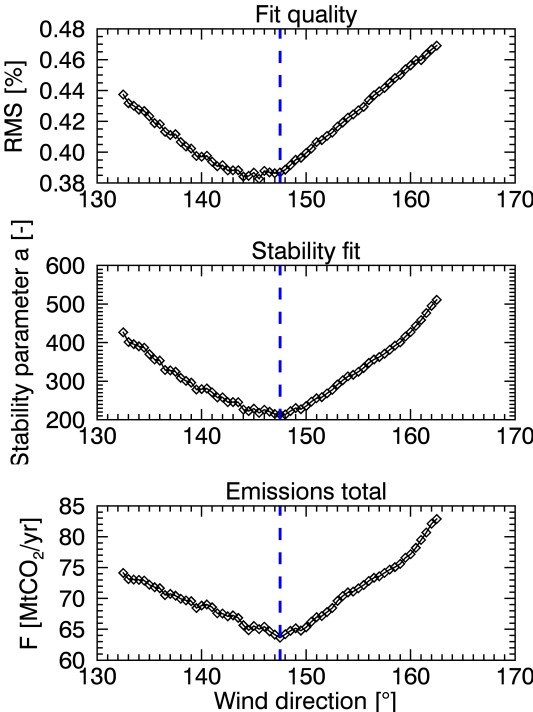

**Figure 11.** Sensitivity of the inverse Gaussian plume model to the wind direction for flux $F$, stability parameter $a$ and the RMS between measurement and fit.

approach. More strict filtering will significantly reduce the data which particularly impacts the mass balance and the defined plume area and was therefore not applied.

The error due to uncertainties in the surface elevation model was not investigated because it largely cancels out for the proxy approach.

5    The overall errors (see Table 2) have been computed as root sum square assuming no correlation between errors and yielding about 10% for the mass balance method (not yet including uncertainties based on variability between results from different flight legs as mentioned above) and 15% for the plume inversion approach.

## 4   Results

### 4.1   In-situ

10    The results of the in-situ flux analysis are shown in Table 3. The average of the six methods of calculation (see Sects. 3.1 and 3.2.2) as the best estimate for the total emission flux is $51.6\,\mathrm{MtCO_2yr^{-1}}$, of which 86% was measured directly.

Emissions for all individual power plants could be derived. For directly measured fluxes (Niederaußem and total cluster in Table 3), the estimates have an uncertainty in the order of 10–20%. Fluxes derived from differences and sums of the primary

**Table 2.** Overview of maximum absolute values of the different error components of the estimated emission rates for the remote sensing results. The uncertainties are derived from a sensitivity analysis and the total error is the root sum square of the individual error components. In case of the mass balance approach an additional uncertainty can be derived from the scatter of inversions for individual tracks (see Fig.15 and Sect. 4.2).

| Error source | Mass balance approach | Inverse plume model |
|---|---|---|
| wind speed uncertainty ($0.5\,\mathrm{m\,s^{-1}}$) | 6% | 6% |
| statistical error (maximum) | * | 6% |
| emission height (200–300 mAGL) | 1% | 0.5% |
| boundary layer depth (1200–1800 mAMSL) | 6% | 6% |
| wind direction (143.5–151.5°) | 4% | 6% |
| source width (100–500 m) | – | 3% |
| grid size (15–55 m) | – | 2% |
| signal filter (11000–13000 counts) | 1% | 6% |
| inclination filter (5–10°) | 1% | 5% |
| $CO_2$ background profile (390 ppm ± 1%) | 1% | 1% |
| Total error | 10% | 15% |

*) statistical errors for the mass balance approach are derived from the resulting emission rates by track in Sect. 4.2

**Table 3.** Summary of all standard methods applied to the different plumes measured in-situ (for more details see the extended table in the supplementary). For the discussion see text in Sect. 3.1. The 'best estimate' is the average of the sensitivity analyses. The minima and maxima were calculated combining the minima and maxima of either the primary measurements (e.g. for the total cluster, Frimmersdorf, and Niederaußem), or by using the worst-case combinations for sums and differences. The split into Neurath old and new was done as shown in Fig. 8.

| Source | best estimate [$MtCO_2yr^{-1}$] | min [$MtCO_2yr^{-1}$] | max [$MtCO_2yr^{-1}$] | error relative to best estimate | fraction of extrapolation | overall uncertainty |
|---|---|---|---|---|---|---|
| total cluster | 51.6 | 50.0 | 61.3 | 11% | 14% | 18% |
| Frimmersdorf | 1.7 | 1.3 | 2.7 | 41% | 45% | 63% |
| Neurath old | 16.8 | 13.3 | 20.8 | 22% | 20% | 32% |
| Neurath new | 7.3 | 3.9 | 11.3 | 50% | 20% | 60% |
| Niederaußem | 25.5 | 22.0 | 26.6 | 9% | 10% | 14% |
| Weisweiler (ideal reference case) | 18.4 | 16.3 | 20.9 | 13% | 0% | 13% |

fluxes have a much higher uncertainty, especially when the plumes are overlapping (Neurath new and old). Frimmersdorf has a high relative uncertainty due to the fact that 45% of flux had to be derived from extrapolation.

## 4.2 Remote sensing

Wind direction was determined from the measured remote sensing data to about 147.5° by both visual inspection and minimiz-
ing the stability parameter (see also Sect. 3.4). When fitting the stability parameter $a$ to the retrieved $XCO_2$ this yields $a$=214.0 ($\pm 8.8\%$ statistical error), i.e. stability class A (very unstable atmospheric conditions, Martin (1976); Masters (1998)), inde-pendent of wind speed. This is in agreement with the observed convective mixing. However, the stability parameter obtained within the inversion is an effective parameter also subsuming other effects such as increased flew gas temperature and variation of wind direction.

Using this stability as input for a vertical Gaussian dispersion model as a function of distance to the source to compute a weighted mean of the wind profile, an average wind speed of $8.2\,\mathrm{m\,s^{-1}}$ was obtained for the plume area covered by the first 9 downwind tracks of the MAMAP data. For this the in-situ calibrated wind model data was used.

Applying this wind speed to the Gaussian plume inversion, the result for the average emission rate for the time of the overflight is in total about $63.6\,\mathrm{MtCO_2yr^{-1}}$ ($\pm 3.0\%$ statistical error) split into $24.0\,\mathrm{MtCO_2yr^{-1}}$ ($\pm 4.6\%$), $14.2\,\mathrm{MtCO_2yr^{-1}}$
($\pm 6.3\%$) and $25.4\,\mathrm{MtCO_2yr^{-1}}$ ($\pm 5.2\%$) for the power plants Niederaußem, Neurath new and Neurath old, respectively. The overall uncertainty including also other components is about 15% (see Sect. 3.4). The contour lines based on the multiple sources as an overlay on the retrieved $XCO_2$ can be seen in Fig. 12. Additionally, data and model result per flight leg are shown in Fig. 13. As mentioned before the evaluated tracks are all located upwind of power plant Frimmersdorf. Therefore no emission rate for Frimmersdorf was derived with this approach.

For the mass balance approach a wind speed was computed for each individual track ranging from about $8\,\mathrm{m\,s^{-1}}$ to $9\,\mathrm{m\,s^{-1}}$. Similar to the inverse plume model, the first 9 downwind tracks were analysed and the associated data are shown in Fig. 13. For three tracks further downwind (see ) where the usual quality filtering could not be applied, the results have to be interpreted with more caution. At this distance, the plume is considerably wider than on the more upwind tracks so that there is less data available for the normalization.

The data were normalised for each flight track individually using a linear fit based on the data outside the plume (see Sect. 3.3.4). The definitions of the outside plume area are listed in Table 4.

The results are shown in Fig. 15 for the individual tracks and the average emission in-between power plants. Figure 15 also shows that there is basically no $CO_2$ influx from upwind into the measurement area. Shown in light grey are the tracks further downwind with decreased data quality. The average absolute deviation from the mean was used as an indicator for the
precision. The precision is worse for cases where emissions are derived as differences which are subject to error propagation from emission estimates from upwind as well as downwind.

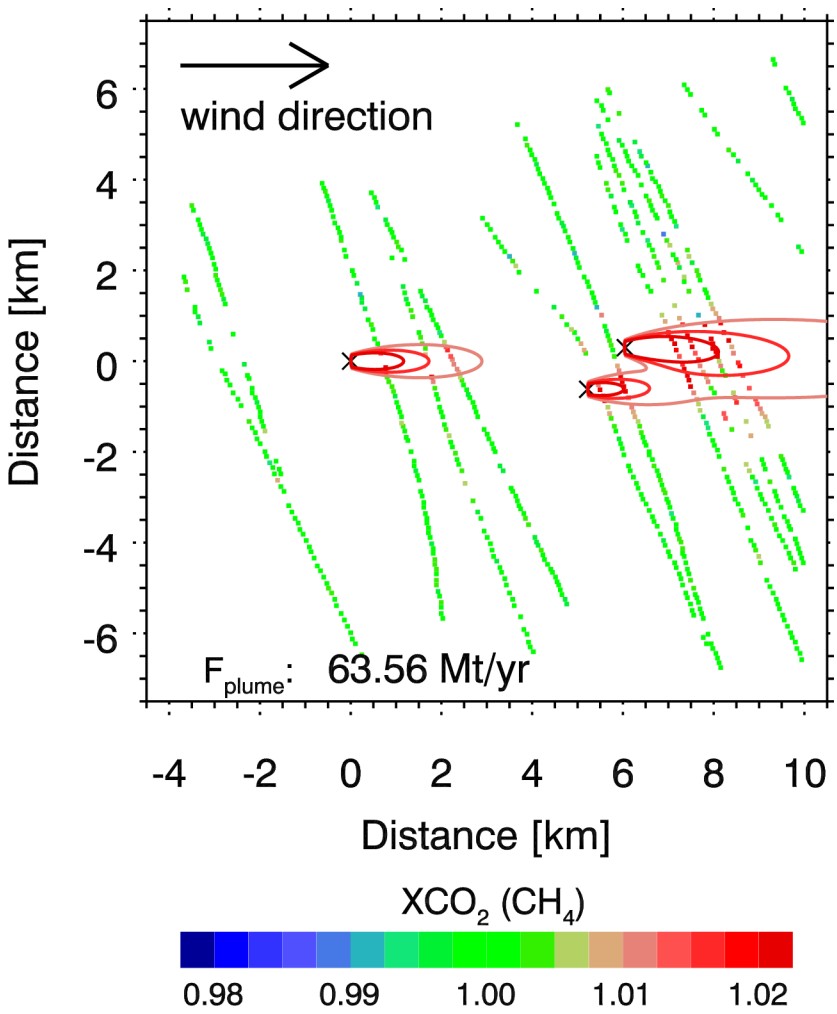

**Figure 12.** Gridded MAMAP XCO$_2$ results rotated so that wind direction points into positive x-direction and contour lines of the inferred plume models for the individual power plants. Total emission rate is $63.6\,\mathrm{MtCO_2yr^{-1}}$ for the time of the overflight. Ground scenes are shown slightly enlarged for better visibility.

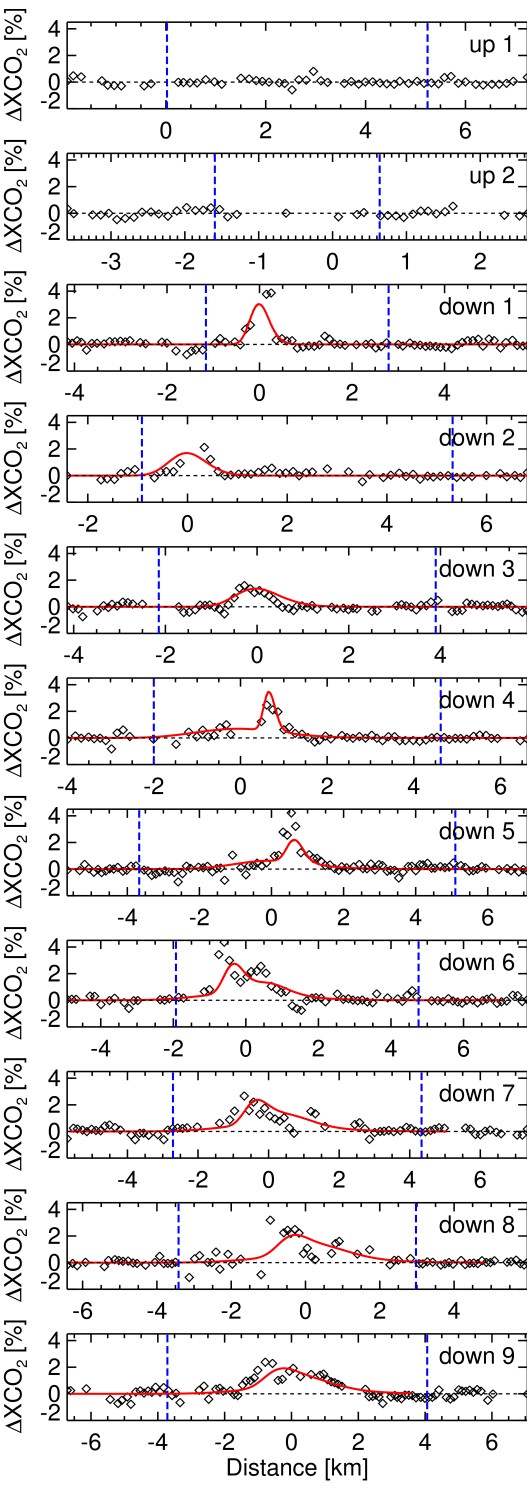

**Figure 13.** MAMAP $XCO_2$ transects for the tracks used for the emission rate estimates (black) and modelled Gaussian plume (red). The areas outside the dashed vertical lines denote the data that were used for the normalisation.

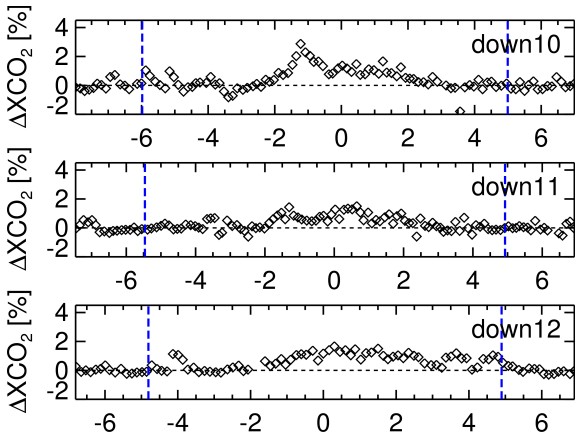

**Figure 14.** MAMAP XCO₂ measurement transects for the 3 flight legs downwind of power plant Frimmersdorf. The area outside the dashed vertical lines denote the data that were used for the normalisation.

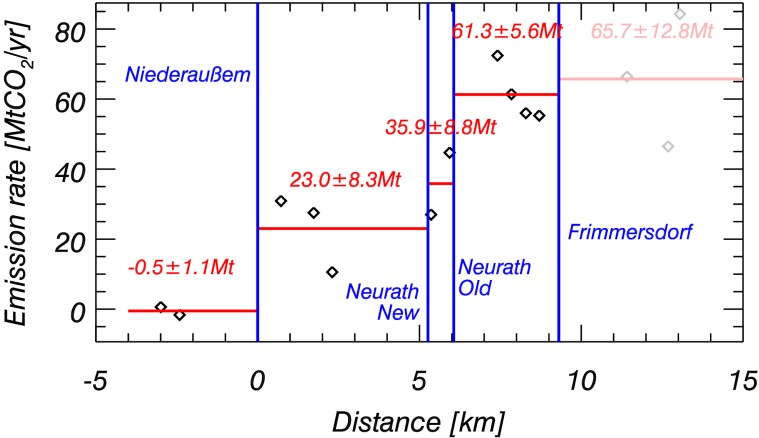

**Figure 15.** Mass balance results based on MAMAP remote sensing data. Vertical lines denote the location of power plants as downwind distance from Niederaußem. Diamonds show emission rates derived from individual aircraft legs, where grey diamonds indicate reduced data quality (see Sect. 3.3.4). Horizontal lines and emission values show average total emissions of the upwind sources.

**Table 4.** Normalisation distances to the end of the measurement track for each individual remote sensing transect.

| Track | Distance to end of track | Comment |
|---|---|---|
| Upwind, Downwind #3 – #5 | 2000 m | Baseline normalisation length for shorter tracks |
| Downwind #1 | 3000 m | Avoid measurements with increase in $CH_4$ next to plume |
| Downwind #2 | 1500 m | Plume not centered on track |
| Downwind #6 – #9 | 3000 m | Track lengths increased |
| Downwind #10 – 12 | 2000 m | Plume is widened due to distance from the source |

## 5 Discussion of results

The $CO_2$ emission rate estimates calculated using the different methods for the different power plants are shown in Fig. 16 and comprise the following: MAMAP remote sensing data analysed with inverse plume model and mass balance approach (Sect. 3.3), in-situ data analysed using the presented mass balance method (Sect. 3.1) and emission rate estimates based on emission factors and energy production data for the time of the overflight. Error bars for the emissions derived from energy production are not shown. The error on power generation itself is generally about 1% (compare also Krings et al., 2011) and the annual error of derived emissions is required to be within 2.5% (European Commission, 2007). The error for the time of the overflight is most likely not much larger. However, a study by Ackerman and Sundquist (2008) shows for individual U.S. power plants that inventories based on monitoring of stack gases and inventories based on emission factors can in principal differ more than 20%.

Generally the two inversion approaches for MAMAP agree very well within their uncertainties for the three individual power plants Niederaußem and Neurath (old and new). However, for the mass balance approach, the uncertainties as determined from the variability of emission estimates for individual power plants (see Fig. 15) are larger when differences were computed due to error propagation. This track by track variability is likely due to instationarity of the atmosphere and shows that repeated measurements are vital to obtain an accurate emission estimate. The inverse plume model, which inverts for all power plants simultaneously, is less affected by the instationarity since all available data is considered for all power plants reducing the overall uncertainty.

When comparing with the $CO_2$ release computed from energy production the agreement is very good for all methods for the emissions from the power plant Niederaußem. For the two Neurath power plants, the remote sensing results indicate less emission from the new units and more from the old units while the overall result is approximately the same. This is then also reflected in the emissions of Niederaußem, Neurath old and new, which is very similar for remote sensing methods and the computed emissions. The combined emissions of the three power plants are 63.6±15%, 61.3±13% und 63.8 $\mathrm{MtCO_2\,yr^{-1}}$ for the MAMAP plume inversion, MAMAP mass balance and the computed emissions. The relative difference to the computed emissions is thereby -0.3% (plume inversion) and -3.9% (mass balance). If no in-situ data had been available, that is if the wind

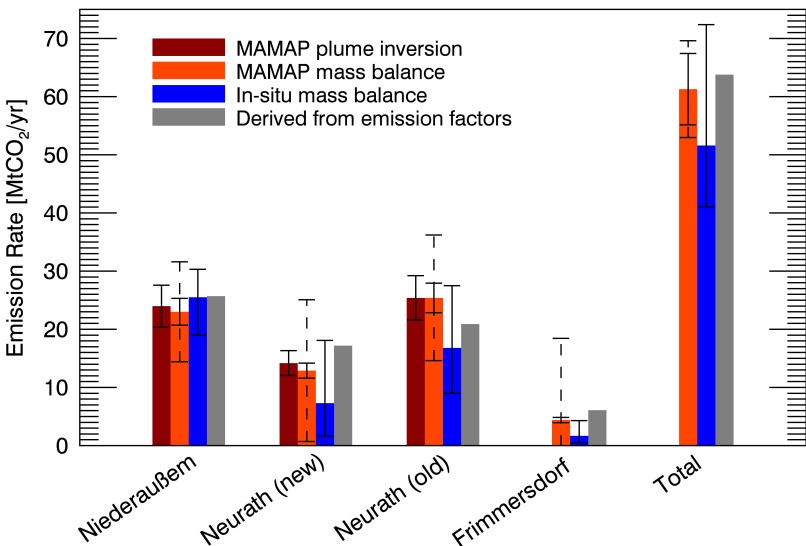

**Figure 16.** Inversion results compared to results obtained from emission factors and energy production for the time of the overflights. For the remote sensing mass balance approach the smaller error bars denote the uncertainties derived from the sensitivity analysis while the larger include also the precision (see Fig. 15) applying the root sum square. The error bars for in-situ are worst-case limits based on the sensitivity analysis and half of the extrapolated emissions below and above of the captured plumes. Emissions from Frimmersdorf were not evaluated with the remote sensing plume inversion.

had been derived only from the COSMO-DE model, the errors would have been -6.4% and -10.0% respectively, reflecting the importance of additional wind measurements.

The total emissions derived from in-situ data are in agreement with the emission computed from emission factors but are somewhat lower. The reasons and uncertainties were explained in Sects. 3.2 and 4.1. However, the selection of the measurement
5   day for detailed analysis was largely driven by the clear sky requirement for remote sensing and there was no ideal overlap between the optimal measurements from the in situ instruments and suitable remote sensing measurement days. Similar for Neurath and Frimmersdorf, while for Niederaußem, where almost the complete plume could be captured the agreement is exceptionally good.

The remote sensing mass balance results for power plant Frimmersdorf are based on data with less strict filters to obtain
10   a sufficient large data set to compute an emission rate. While the scatter and, hence, the uncertainty is quite large, the mean value indicates an emission rate of $4.4\,\mathrm{MtCO_2yr^{-1}}$ which is close to the emission rate of $6.1\,\mathrm{MtCO_2yr^{-1}}$ based on emission factors. However, the associated error based on the sensitivity analysis and precision is about $8.3\,\mathrm{MtCO_2yr^{-1}}$.

# 6 Conclusions

This work enhances the comparison between measurement and inversion approaches using in-situ and remote sensing data to obtain emission rates for flue gases from a cluster of point sources with known locations. These sources were partly in close proximity to each other and the plumes of – in this case – $CO_2$ from coal fired power plants overlapped adding complexity to the inverse problem.

In contrast to the in-situ method, the remote sensing measurements required clear sky conditions at the time of the measurement. MAMAP measures solar backscattered electromagnetic radiation in the short-wave infrared. To simplify the radiative transfer calculations, cloud free atmospheres are generally selected to avoid the radiative transfer issues associated with solar electromagnetic radiation passing through clouds. The selection of the measurement day for this study was largely driven by this requirement. This restriction impacted the selection of the measurement day for the analysis in this work. This resulted in some days with potentially more favorable conditions for the in-situ method (coverage, flight restrictions, etc.) being disregarded. Nevertheless, the in-situ measurements for the selected day allowed a good estimate of the emission rate when the extrapolation up to the limiting stable layer was applied.

Both remote sensing point source inversion methods are able to quantify the emissions within the error bars – about 15% for the plume inversion approach and about 13% for the mass balance method referring to the combination of the three power plants. The mass balance approach requires less parameters. It is, for example, not essential to know the exact source location and dimensions which is an advantage for surveying unknown sources with a non-imaging instrument like MAMAP. However, the mass balance approach showed a lower precision when only few flight legs per source are available, in particular close to the source and when differences between inversion results are interpreted. To mitigate this effect which is likely based on atmospheric instationarity it is of advantage to gather measurements on multiple flight legs. This results in a higher precision and an improved error estimate.

One critical external input parameter for the analysis of the remote sensing data is wind information, which in this work was derived using model and in-situ data. While the wind direction can be fitted to the data directly, this is not possible for the wind speed which scales linearly with the emission rate.

The in-situ inversion proves to be accurate for power plant Niederaußem where an almost complete sampling of the plume was possible. Further away from the sources, capturing the complete vertical plume extent in the higher reaching convective boundary layer was not possible due to airspace restrictions.

While the individual results for remote sensing and in-situ yield very similar results provided sufficient sampling, a joint inversion approach may complement the individual methods also when there is no complete plume coverage. The accuracy of in-situ error estimates for cases with better coverage in less restricted places is better than 15% (reference case in Table 3).

The methods presented here are demonstrated for $CO_2$ emissions from point sources, however, they are directly applicable in the same way to other gaseous compounds that disperse in the atmosphere and that have a lifetime longer than between emission and measurement, such as, for example, $CH_4$ which can also be derived from MAMAP remote sensing observations.

This case study illustrates the advantages and disadvantages of the used methods. The remote sensing approach needs clear sky conditions but offers the possibility to perform many flight legs in a short period of time. This is necessary to reduce the uncertainty as can also be seen from Fig. 15. The multiple transects allow for the application of the Gaussian plume model to a multi source setup which simultaneously retrieves the emission rates from several sources.

While for MAMAP the plume model usually utilises a priori information on the source location, an imaging instrument with sufficient spatial resolution and sensitivity (similar to, for example, AVIRIS-NG (Thompson et al., 2015; Frankenberg et al., 2016), though having a lower sensitivity compared to MAMAP) is able to determine the source location from the data directly and can acquire more data on shorter time scales potentially reducing uncertainties on derived emission estimates. Furthermore imaging instruments offer the possibility of mapping large areas in a survey for unknown sources.

However, there is generally the need for external wind information which originates from models and/or in-situ measurements. The analysis in this study shows an overestimated model wind speed of about 6% (or about $0.4\,\mathrm{m\,s^{-1}}$) which is smaller than the uncertainty on wind speed. So in this case relying on the model alone may be sufficient. In a former study of similar setup (Krings et al., 2013) the error was about 10% or ($0.7\,\mathrm{m\,s^{-1}}$). A wider and systematic analysis on the accuracy of the model wind is needed to assess to what extent additional wind (profile) measurements are dispensable. This will also become more relevant with regard to observations of localized sources by current and upcoming satellite missions with increased accuracy and spatial resolution. In these cases additional wind measurements will generally not be available.

In contrast to the remote sensing measurements of the entire vertical column, in-situ measurements need to sample the plume with flight legs at different altitude levels. As a result of the time needed to complete a representative vertical cross section of measurements, only a limited number of repeated measurements are typically feasible. Interpolations within the cross sections and extrapolations to the surface and sometimes to the top of the plume have to be applied. This also applies for this study, where the boundary layer reached into restricted airspace. However, the in-situ method has the advantage of delivering vertically and horizontally resolved information in conjunction with co-located wind information, which can be readily used to infer a flux estimate. The high intrinsic sensitivity enables the detection of elevated trace gas levels also at great distances to the source. Errors on the inversion results from in-situ and remote sensing data are rather similar.

*Acknowledgements.* The measurement campaign C-MAPExp was funded by the European Space Agency (ESA). The MAMAP activities are funded in parts by the University of Bremen and the State of Bremen. Wind data from the COSMO-DE model were obtained from the German Weather Service (DWD). The reference values for $CO_2$ emission rates of the power plants were derived from power generation values of each power plant location kindly provided by RWE AG, Essen, Germany. We acknowledge Armin Jordan from the Max Planck Institute for Biogeochemistry in Jena, Germany, for the analysis of airborne grab samples at the BGC GasLab. Finally, we would like to thank Christian Frankenberg, the anonymous referees and Levi Golston for providing detailed comments that helped to improve this study.

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
