# Peer review of "Airborne remote sensing and in-situ measurements of atmospheric $CO_2$ to quantify point source emissions"

_Atmospheric Measurement Techniques, 2016_

## Referee Comment (RC1) · Anonymous Referee #1 · 31 Dec 2016

This paper describes an attempt to estimate CO2 emissions from coal-burning power plants in Germany using either airborne remote sensing measurements or airborne in-situ sampling within the plume. The data analysis and emission estimates use either a mass-balance approach or a Gaussian plume assumption. The uncertainty analysis leads to relative errors on the order of 10-15% and the estimate comparison to reported emissions are consistent with this relative errors. This paper contains a lot of interesting material for an evaluation of the potential and difficulty of such approaches for the estimate of CO2 emissions from point sources. It should eventually be published. On the other hand, I have been very disappointed by the manuscript presentation that looks more like a experiment report than a scientific paper. Also, the paper lacks

conciseness and many figures are not necessary. Although the paper combines the in-situ and the remote-sensing approaches, there is no real discussion of the pros and cons of both. Although the in-situ wind speeds are used for the interpretation of the remote sensing data, the in-situ concentration observations (Figure 5) should have been used, I think, for a discussion on the validity of the Gaussian plume model. It is really not clear why the emission from the Frimmersdorf power plant could not be estimated with the remote sensing technique (P17-18). Indeed, there are many flight tracks downwind of this power plant that, in principle, could be used for that purpose. I assume that the authors have attempted an inversion, with no success, so that they chose to discard this estimate. Their experience on that particular aspect should be clearly stated to help in the design of future similar campaigns. Perhaps only flight tracks within a few kilometers from the emission can be used ?

The paper is strangely organised. The method for the remote sensing approach is mostly described in the "Result" section. Besides, it is rather strange to have in situ measurements, such as Figure 10, presented in the Remote Sensing section rather than the In situ section. There is a need to show early in the manuscript (section 3) the flight track (both in-situ and remote sensing), similar to Figure 12, as well as the location of the "virtual wall" that was chosen for the mass balance estimate

In the "wall" approach (Figure 4), I could not understand why several cells are considered in the along wind direction. Why not assume that the cell dimensions are zres (vertical) x hres (cross wind) x d (along wind). I could not understand the discussion on page 7 lines 27-30. The dimension of the wall is not provided.

Detailed comments: P2L15: Are you suggesting that thermal infrared observations provide valid concentration estimates in the presence of clouds ? P4L12 : Please provide a valid argumentation why the method used in the manuscript is better than krigging P7L16-22. Not clear why there is a need to have the virtual wall oriented precisely crosswind. It seems more important to have the wall aligned with the flight tracks P8L9: Could not understand P9L2-3 : Could not understand P9L6: What about

the sampling of the plume and its variability ? What is the variability of the concentration with the wall cells? P9L15: It is rather difficult to understand that there is an uncertainty about the top of the mixing layer, but not on the flux close to the surface. Either one assumes that there is little vertical mixing, in which case only the flight track at a level close to that of the chimney, or there is mixing that transfers $CO_2$ both high in the mixing layer and towards the surface (see P9L30). P9L20: Instationarity of the source is mentioned. What is the variability of the source according to the power plant management ? P10L5-10. Although it is not stated clearly, I understand that the discussion is for various days. The paper should rather provide the result for the particular day that is analysed in the manuscript, and make a single sentence for the other days. P10L30 "to the top of the well mixed boundary layer". In situ measurements shown in the manuscript (Figures 4 and 5) clearly show that the boundary layer is not "well mixed' P12L1. Although section 5.2 is supposed to show "results", it actually mostly describes the method. P12L24. Justification not clear. 0.9% relative to what ? P12L26: There is no justification for the removal of data "close to saturation". As long as there is no saturation, these data should have a high SNR. Please justify P14L1. It is said that the elevated $XCO_2$ are well aligned with the wind field from the power plants, but it seems to me that the high value are further North-East that what would be expected P14L6: It is said that the boundary layer depth is important to compute the wind field. However, the in situ measurements clearly show that the boundary layer is not well mixed. P15L1 : Figure 11 shows that there was a significant decrease of the wind speed during the time of the in situ measurements. This should affect the intensity of the plume and I am surprised this was not discussed in the in-situ section. P17L4. Are there realy any significant difference for the modelled wind speed over the 10 km area ? P17L8. Section 5.2.4 is supposed to be a "Result" section. Yet, a large fraction of it describes the method P17L16: It is said that the measurements are P18L4. Description of the wind speed estimate. It is said that a Gaussian profile for the concentration is assumed. Yet, the in situ measurement do not show such Gaussian profile. It would be nice to compare the assumed vertical distribution of $CO_2$ with the in situ measurements. Also, the fact that there is little vertical gradient in the wind speed makes this discussion somewhat unnecessary. How is done the weighting to derive a mean wind speed ? P18L18. "Very unstable atmospheric conditions". Is that consistent with the observed meteorological conditions on that day ? P23L22: The authors state the error analysis leads to an uncertainty on the order of 10% for the mass balance approach. This is in contradiction, I believe, with the results shown in Figure 15 that show larger variations for the various leg estimates. For instance, three legs can be used to estimate the emissions from Niederaussem. There is a factor of 2 between the largest and the smallest. This appears contradictory with the error analysis, in particular since several of the error sources are biases and cannot explain a difference between the estimates from two nearby legs. I am surprised this is never discusses in the text. P21L1: The whole section 5.2.5 is poorly written. P24L4: "can differ more than 20% for individual power plants". So what are you saying here ? Are you suggesting that the reported emissions shown in the paper (Figure 18) can be off by that much ?

In the following, I make comments on the figures. I strongly believe that several of them are not useful whereas other could bring additional information Figure 1 : Figure 2 : Limited usefulness Figure 4: Provide colour scale Figure 5 : Is this figure supposed to show the same data as in Figure 4 ? I cannot recognize any feature. I strongly recommend to show the value of the measurements within the circles that are used for the interpolations. What is the link between this figure and the "wall" approach shown in Figure 4 ? Figure 6: Definitely not useful. Not clear what is really shown (ie RMS of what, relative to what ?) Figure 7: Definitely not useful Figure 8: Is it really XCH4 as indicated in the legend, or XCO2 ? Why no color scale ? Figure 9: Marginaly usefull. The text could simply say that the in situ measurements (potential temperature and aerosol) provide no useful information to determine the top of the boundary layer up to 1100 m Figure 10 : Should definitely be presented in the "in situ" section, together with Figure 5, and not in the remote sensing section. Figure 11: Difficult to read. Values for the X-axis could be simpler (e.g. 5/10/15) Figure 12: Should be shown early on in manuscript. Why is the color scale not adjusted to the data (no observation before 12)

Figure 13: I suggest to reduce the range of the color bar to 0.99-1.02 and have color lines (Gaussian plume model) for 1.005, 1.01, 1.015 and 1.02 Figure 14: I strongly suggest to add, on each of these graphs, a line showing the result of the modeling according to the Gaussian plume approach. Also, add a horizontal line to show the 0. Figure 15 : State explicitly in the legend that each symbol corresponds to a flux estimate derived from a given aircraft leg. Figure 16 : Not useful Figure 17 : Not useful

[Figure]

---

## Short Comment (SC1) · 22 Jan 2017

The authors describe two methods for quantifying point source emissions, one based on in-situ mass balancing and one on remote sensing coupled with an inverse Gaussian plume model, along with field data comparing the two methods. I would like to add three comments on the description of the in-situ method, which I hope will help strengthen that section.

It was surprising that Gordon et al. 2015 (AMT 8:3745-3765) and Cambaliza et al. 2014 (ACP 14:9029-9050) are not discussed or even cited, given that they both investigate in detail the uncertainties of the in-situ aircraft mass balance methodology. Gordon et al. 2015 discusses issues of interpolation, extrapolation, turbulent fluxes,

and issues related to determining the background concentration in the context of determining emissions from an elevated source as was done here, while Cambalizia et al. 2014 also considers interpolation, boundary layer entrainment, and other effects. On page 4 transparency of interpolations and extrapolations is claimed as a benefit here, but seem less well developed than in either of those papers. The description of both on Page 8, is actually specifically not transparent and neither seems to be included in the error budget of Table 4. Part of the reason why the in-situ method is discussed in such detail seems to be because a variation on the mass balance method is presented, however the differences and benefits are not clearly distinguished in Section 4.1 or the results.

I also recalled that Figs 3, 4, and part of 2 are identical to Figure 3 in Hacker et al. 2016 (Animal Production Science 56:190-203), who cited the report from the authors of the current paper, Bovensmann et al. (2014). Since the figures are now also in Hacker et al., I think that the original Bovensmann et al. (2014) should be referenced here to avoid confusion.

Finally, it would be helpful to know whether the turbulent (5 Hz) could be resolved as indicated in the caption for Figure 3 or if there was attenuation, and how the inclusion of the turbulent flux compares to not including it.

---

## Referee Comment (RC2) · C. Frankenberg (Referee) · 31 Jan 2017

The manuscript "Airborne remote sensing and in-situ measurements of atmospheric CO2 to quantify point source emissions" by Krings et al presents results from an airborne campaign, inferring point source fluxes of CO2 using both mass balance approaches as well as a Gaussian plume modeling of remotely sensed total column averages. Even though the data presented here is indeed interesting, I tend to agree with reviewer 1 that it often reads much like a report and would need restructuring and more concise (and precise!) language. At times, the authors get caught up in details that are not entirely relevant to the study at hand, e.g. Figs 9-12 are too detailed or not necessary (9-10) or misplaced in the respective section. I would suggest putting a

general description of the domain as well as the data right at the beginning (e.g. showing MAMAP footprints as well as in-situ ground projections on the map in Figure 1. It would greatly help setting the stage for the discussion.

Some more specific comments: Abstract last sentence: this is a sudden topic break and needs some rephrasing

Page 3, line 30: wheras (there are many small things like this or "straight forward", which is one word. I won't go into more details, the copy-editor should catch those at a later stage but some sentences are too literal translations from german.

Section 4: I think this is poorly described and justified and I urge the authors to consider revisiting the differences between their approach and the paper Levi Golston mentioned.

E.g.

I) you mention "Kriging" is not necessarily the best suitable approach. If you provide critic, you have to back it up either with an analysis or a citation. There is also no real explanation what kind of interpolation schemes you are using (apart from the boundary voxels).

II) You mention that you include turbulent fluxes. This is very interesting and I was excited but then I didn't see any further analysis. Did you compute the differences with or without turbulent components? What is the relative error in your case? Can you plot c-mean(c) vs v-mean(v) for some voxels to show the correlations as expected for turbulent fluxes?

III) Wind speed seems to be a dominant error, do the others actually matter? You will need to provide realistic estimates regarding kriging and turbulent fluxes, otherwise the reader won't be able to judge the importance (even though this specific case study might not lend itself to extrapolation to a general case). Page 5, line 9: As above, please show what error you incur by doing mean(v)*mean(c) vs. mean(v*c)

Figure 4: Please add color-bar and make this a realistic example based on real data. How many data points to average do you typically have per voxel?

Page 9, line 8: Conditio sine qua non: Even though I have a "Grosses Latinum", I had to look it up again. Please rephrase in plain english, esp. as it is here used in a rather trivial way, not warranting the grandiose latin phrase ;-). One might argue though that precision "could" be important if it is really bad while accuracy won't matter. This could be a factor when flying very cheap instruments on small unmanned aerial vehicles near the plume. So I would keep the discussions as general as possible.

Page 9, line 17: Chimneys: It would be good to discuss how well you could measure the fluxes if the emissions were to happen at the surface. What would this imply for the in-situ based approach and potential flight-paths.

Figure 5: This figure confuses me. I "assume" the dots are actual measurement locations. Given what you wrote, there is a constant extrapolation to the surface. However, it doesn't look that way for the second little intrusion at x=0. Also, there are a couple of local maxima in between dots. You need to explain the interpolation scheme and this would a good place to compare against kriging or other interpolation schemes. Also, if the dots are measurements, please color-code them with the actual measurement values at that x-y position. This will help evaluate the interpolated fields better. A last point: Why is this continuous on x and y? Wouldn't it make sense to sketch out the actual grid boxes here as well? Page 12, line 3: Please add citation for proxy method (this is not common knowledge).

Figure 6: I think the figure itself doesn't tell more than the text, could be skipped. Same with Fig. 7

Page 14, line 17: So in essence, you don't really need the wind speeds in this case as the error is rather small?! Ideally, you won't always need both aircraft. If there is confidence in modeled winds, it would be a good sign for future remote-sensing only campaigns.

Figure 10: Weird x-spacing (value 493?). also better to use same x-scale for both subplots.

Page 21, line 10 +/-: Wouldn't you ideally fit a Gaussian model with a vertical wind-speed profile? This would rather directly model the total column AND the wind-profile. How high is your Gaussian profile extending to the vertical? That might be a plot to add (or is it just 2D in x and y?).

Page 23, line 19: "This is because they to a good extent cancel out..." "largely" cancel out?

Page 26, line 20: I think they gase don't need to be inert, just have lifetimes much longer than the time between emission and measurement. I would guess even NOx emissions could work with an "inert" assumptions on this very local scale.

As a last general point: Please try to re-structure somewhat to bring out the key messages in a more concise way (and illustrate better how your in-situ inversions differ from others). At the end, provide a more generic overview of both flux estimates and its pro/cons and path forward. This could extend to a discussion using high-resolution mapping like the cited AVIRIS-NG papers (which should be cited at page 26, line 9).

Last but not least my sincere apologies for the late review.

---

## Author Comment (AC1) · 30 Sep 2017

**Reviewer 1:**

We would like to thank the reviewer for the helpful and detailed comments. In the following we carefully address the comments point by point. First we repeat the reviewer's comment (R#...) and then give our response (A).

**R#1: This paper describes an attempt to estimate CO2 emissions from coal-burning power plants in Germany using either airborne remote sensing measurements or airborne in-situ sampling within the plume. The data analysis and emission estimates use either a mass-balance approach or a Gaussian plume assumption. The uncertainty analysis leads to relative errors on the order of 10-15% and the estimate comparison to reported emissions are consistent with this relative errors. This paper contains a lot of interesting material for an evaluation of the potential and difficulty of such approaches for the estimate of CO2 emissions from point sources. It should eventually be published. On the other hand, I have been very disappointed by the manuscript presentation that looks more like a experiment report than a scientific paper. Also, the paper lacks AMTD Interactive comment Printer-friendly version Discussion paper conciseness and many figures are not necessary. Although the paper combines the in-situ and the remote-sensing approaches, there is no real discussion of the pros and cons of both.**

A: We added a discussion on the pros and cons of the respective methods in the revised manuscript:

*This case study illustrates the advantages and disadvantages of the used methods. The remote sensing approach offers the possibility to perform many flight legs in a short period of time. This is necessary to reduce the uncertainty as can also be seen from Fig. 14. The multiple transects allow for the application of the Gaussian plume model to a multi source setup which simultaneously retrieves the emission rates from several sources.*

*While for MAMAP the plume model usually utilises a priori information on the source location, an imaging instrument with sufficient spatial resolution and sensitivity (similar to, for example, AVIRIS-NG (Thompson et al., 2015; Frankenberg et al., 2016), though having a lower sensitivity compared to MAMAP) is able to determine the source location from the data directly and can acquire more data on shorter time scales potentially reducing uncertainties on derived emission estimates. Furthermore imaging instruments offer the possibility of mapping large areas in a survey for unknown sources.*

*However, there is generally the need for wind information which originates from models and/or in-situ measurements. The analysis in this study shows an overestimated model wind speed of about 6% (or about 0.4m/s) which is smaller than the uncertainty on wind speed. So in this case relying on the model alone may be sufficient. In a former study of similar setup (Krings et al., 2013) the error was about 10% or (0.7m/s). A wider and systematic analysis on the accuracy of the model wind is needed to assess to what extent additional wind (profile) measurements are dispensable. This will also become more relevant with regard to observations of localized sources by current and upcoming satellite missions with increased accuracy and spatial resolution. In these cases additional wind measurements will generally not be available.*

*The remote sensing instrument MAMAP measures solar backscattered electromagnetic radiation in the short-wave infrared. To simplify the radiative transfer calculations, cloud free atmospheres are selected to avoid the radiative transfer issues associated with solar*

*electromagnetic radiation passing through clouds. The selection of the measurement day for this study was largely driven by this requirement. This generally involves a more convective and therefore thicker boundary layer making the gathering and analysis of the in-situ measurements more complex.*

*In contrast to the remote sensing measurements of the entire vertical column, in-situ measurements need to sample the plume with flight legs at different altitude levels. As a result of the time needed to complete a representative vertical cross section of measurements, only a limited number of repeated measurements are typically feasible. Interpolations within the cross sections and extrapolations to the surface and sometimes to the top of the plume have to be applied. This also applies for this study, where the boundary layer reached into restricted airspace. However, the in-situ method has the advantage of delivering vertically and horizontally resolved information in conjunction with co-located wind information, which can be readily used to infer a flux estimate. The high intrinsic sensitivity enables the detection of elevated trace gas levels also at great distances to the source. Errors on the inversion results from in-situ and remote sensing data are rather similar.*

**R#2: Although the in-situ wind speeds are used for the interpretation of the remote sensing data, the in-situ concentration observations (Figure 5) should have been used, I think, for a discussion on the validity of the Gaussian plume model.**

A: The Gaussian shape of the vertical distribution can only be observed close to the source. Due to the "reflection" of the plume off the surface and off the top of the boundary layer, the $CO_2$ gets mixed rather rapidly. About 2 km downwind of the source the $CO_2$ is well mixed according to the Gaussian model, i.e. on average and not necessarily for any snapshot in time. The in-situ measurements, for example, downwind of power plant Niederaußem (Fig. 5) are more than 2 km downwind of the source so that no distinct Gaussian shape in the vertical concentration profile is to be expected.

For better assessment of validity of the horizontal plume model, on the other hand, which is used to infer the emissions, the model result is now shown in addition to the data for the individual remote sensing flight legs (see also R#48).

**R#3: It is really not clear why the emission from the Frimmersdorf power plant could not be estimated with the remote sensing technique (P17-18). Indeed, there are many flight tracks downwind of this power plant that, in principle, could be used for that purpose. I assume that the authors have attempted an inversion, with no success, so that they chose to discard this estimate. Their experience on that particular aspect should be clearly stated to help in the design of future similar campaigns. Perhaps only flight tracks within a few kilometers from the emission can be used?**

A: In this study, we previously discarded the flight tracks further downwind due to large data gaps as stated in the main text (p17 L17 – p18 L3). In addition, the measured concentrations downwind of Frimmersdorf are an integrated composite of all sources upwind. For example, the power plant Niederaußem is more than 10 km upwind of these tracks so that part of the enhancements might already be dispersed to the flanks of the transects which we require for normalization – as mentioned in the text.

However, as the reviewer encourages us to investigate the measurements from further downwind, we relaxed the signal threshold for the first three tracks downwind of power plant Frimmersdorf to a minimum of 3000 counts and the inclination filter to 15°. In this way we ensure that a sufficient set of measurements, even if of lower quality, are available for interpretation. The mass balance flux estimates for these tracks are shown in an updated result

plot. Although the result shows some scatter, the average is reasonable. Since we did not apply our usual quality filter, these results have to be interpreted with more caution.

We did not apply the Gaussian plume method for these data as that would require to mix data which were subject to different filter criteria.

A brief discussion has been added to the revised manuscript.

**R#4: The paper is strangely organised. The method for the remote sensing approach is mostly described in the "Result" section. Besides, it is rather strange to have in situ measurements, such as Figure 10, presented in the Remote Sensing section rather than the In situ section.**

A: Agreed, the description of the remote sensing method was moved to Section 4. The original reason to have Figure 10 in the remote sensing section was that it is specifically referred to here. The Figure was removed for the revised version. Figure 11, however, was left in the remote sensing section. Although it shows in-situ data, the plot specifically addresses the remote sensing analysis.

**R#5: There is a need to show early in the manuscript (section 3) the flight track (both in-situ and remote sensing), similar to Figure 12, as well as the location of the "virtual wall" that was chosen for the mass balance estimate**

A: The new figures 1 to 3 are providing this now in a clear manner. The concept of the 'wall' was abandoned. We describe it now as cross-sections with maximum distances from which a projection along the wind was allowed. The concept is the same, but, in other words. This point is commented later again.

**R#6: In the "wall" approach (Figure 4), I could not understand why several cells are considered in the along wind direction. Why not assume that the cell dimensions are zres (vertical) x hres (cross wind) x d (along wind).**

A: As mentioned in the answer to R#5, the wording has changed. Along with this are the cross-sections in the new Figures 3, and 4 to 8. All these examples are with real data, and not anymore a schematic figure like Fig. 4, that was obviously confusing.

**R#7: I could not understand the discussion on page 7 lines 27-30. The dimension of the wall is not provided.**

A: See answer to R#6.

**R#8: Detailed comments: P2L15: Are you suggesting that thermal infrared observations provide valid concentration estimates in the presence of clouds?**

A: Yes, as long as the clouds are not in between target and instrument optics. However interpretation of thermal infrared measurements depends on the thermal contrast, as there is no signal if, for example, the $CO_2$ has the same temperature as the surface (Young, 2002).

In contrast, short-wave infrared observations as used in this study require backscattered sunlight. Clouds may block the solar electromagnetic radiation or increase the radiative transfer complexity in the determination of the path of the electromagnetic radiation due to (multiple) scattering.

**R#9: P4L12 : Please provide a valid argumentation why the method used in the manuscript is better than krigging**

A: As a main part of the revision, we did the whole calculations with our linear inter- and extrapolation method with only four rules, and with Kriging. There are two aspects to distinguish: (i) about the inter- and extrapolation. By using Kriging as another method we have shown that the difference is small. (ii) More important seems to be averaging and interpolation of fluxes, instead of averaging mass- and wind-fields before calculating the fluxes. Especially when the latter was done by Kriging, the deviation from the ensemble of other solutions is increasing, most likely due to the fact, that small artefacts in the individual fields are increasing the errors. Bottom line: Our method is not better than Kriging, but, we should not regard Kriging as the only option. This is also true after studying Gordon et al. (2015) in detail. We think that the new text is much clearer in showing and discussing these details. Since a complete revision was performed, and the separation of more individual sources was possible, the results as displayed in table 3 and Figure 18 were updated. The details are presented in a separate supplement

**R#10: P7L16-22. Not clear why there is a need to have the virtual wall oriented precisely crosswind. It seems more important to have the wall aligned with the flight tracks**

A: We disagree with. Aligning with the flight track is possible as long as there is only one track, or several perfectly stacked above each other. Then it does not make a difference. However, in a real case, with a flight pattern that was not ideal by several reasons, it is very important to have cross sections exactly perpendicular to the wind during the time of observations, because otherwise, maxima on different flight tracks would add in different grid cells. This can be avoided when the projection to the cross-sections (we do not call it 'wall' anymore) is along the wind, to a perpendicular plane. The new figures 2 and 3 should explain this.

**R#11: P8L9: Could not understand**

A: Should now be clear with the new explanations about the inter- and extrapolations, and the percentage of directly measured fluxes in relation to the extrapolations below and above the flight tracks.

**R#12: P9L2-3 : Could not understand**

A: When the (systematic) error in the wind speed measurement is 0.5 m/s, this would modify the total flux in a 5 m/s wind by 10%. It is less if the error is non-systematic. However, this is a worst-case estimate. With the same 0.5 m/s in error, a flux in a 10 m/s would be wrong by 5% only, but, by 25% in a flow of 2 m/s. On the other hand, 0.5 ppm error would only contribute an error of 1% for the flux in a typical moderate plume with 50 ppm enhanced $CO_2$. We argue here, that under the conditions of these measurements (plume enhancements usually higher than 50 ppm) and wind speeds around or above 5 m/s, the errors of the measurements (instrumental errors, both systematic and stochastic) are contributing a maximum of 10 % and that the wind is more critical than the concentrations. This finding is well in agreement with Gordon et al. (2015).

**R#13: P9L6: What about the sampling of the plume and its variability ? What is the variability of the concentration with the wall cells?**

A: This is now clearly shown in Figures 4 to 8, and with the initial data in Figures 2 and 3.

**R#14: P9L15: It is rather difficult to understand that there is an uncertainty about the top of the mixing layer, but not on the flux close to the surface. Either one assumes that there is little vertical mixing, in which case only the flight track at a level close to that of the chimney, or there is mixing that transfers CO2 both high in the mixing layer and towards the surface (see P9L30).**

A: Primarily we show that the amount of fluxes coming from the extrapolation is only 10, and 14% in the two budgets that were directly measured (more in those that were derived as sums or differences). The different methods are discussed: Fluxes or concentrations staying constant or diminishing to zero above background, etc. Finally we attributed half of the extrapolated amounts to the overall error (Table 3), which means that the extrapolation has an uncertainty of 50%. More details are discussed in the revised text and should be clearer from the new figures.

**R#15: P9L20: Instationarity of the source is mentioned. What is the variability of the source according to the power plant management ?**

A: We added the information to the revised version. For this case study, the source variation based on energy production was less than 0.5% for Niederaußem, Neurath new and old blocks. For Frimmersdorf the variability was about 4% but with considerably lower total fluxes.

**R#16: P10L5-10. Although it is not stated clearly, I understand that the discussion is for various days. The paper should rather provide the result for the particular day that is analysed in the manuscript, and make a single sentence for the other days.**

A: Except for the reference case of power plant Weisweiler in the new Table 3, all results and discussions are about this specific day. This is clearer now in the revised version.

**R#17: P10L30 "to the top of the well mixed boundary layer". In situ measurements shown in the manuscript (Figures 4 and 5) clearly show that the boundary layer is not "well mixed'**

A: We agree, that "well mixed" was not the best expression to describe the situation within the plume relatively close to the sources. We do not use it anymore in this context. However, it applies for the boundary layer in terms of water vapor or aerosols on the regional scale, enabling us to estimate the top of the actual convective boundary layer. Convective dispersion was evidently acting within this layer below 1300 mAMSL. Please also note that Fig. 4 was conceptual (now replaced). The heterogeneity of the plumes is clearly stated, and is the prime reason for the method applied for the inter- and extrapolation.

**R#18: P12L1. Although section 5.2 is supposed to show "results", it actually mostly describes the method.**

A: Agreed, this section was moved to the methods chapter.

**R#19: P12L24. Justification not clear. 0.9% relative to what ?**

A: The RMS is relative to the model: RMS[(model-measurement)/model] where the choice of 0.9% follows from Figure 6. This is where a strong decrease in fit quality begins. However, also from Figure 6 it can be gathered that not many data is affected by this. As the reviewer is of the opinion Figure 6 is "definitely not useful" (see below) we removed the Figure and added following lines to the revised version:
*Filtering, based on the spectroscopic fit quality, has been applied rejecting measurements with a root mean square (RMS) value of the differences between measurement and model after the fit larger than 0.9% relative to the model affecting about 0.1% of the total measurements. The threshold was empirically determined from the distribution of RMS values ordered by size (compare also Krings et al., 2011, 2013).*

**R#20: P12L26: There is no justification for the removal of data "close to saturation". As long as there is no saturation, these data should have a high SNR. Please justify**

A: We added more information to the revised version:
*Filtering of the data accounts for not only SNR but also whether linear full well is achieved. For the full well ADC range chosen by the manufacturer a non-linear behavior could be observed for very high detector fillings. Therefore data with very high filling factors are excluded from further processing. However, out of all measurements, the chosen maximum threshold value affects only 4 single measurements (all in one burst) during the whole measurement period.*

**R#21: P14L1. It is said that the elevated XCO2 are well aligned with the wind field from the power plants, but it seems to me that the high value are further North-East that what would be expected**

A: Considering the complexity of the atmosphere (turbulence or puffiness of air masses with high $CO_2$ concentrations) we consider the average alignment of the overall plume structure of all power plant emission with the determined wind direction to be quite good as can also be seen from Figure 13.

**R#22: P14L6: It is said that the boundary layer depth is important to compute the wind field. However, the in situ measurements clearly show that the boundary layer is not well mixed.**

A: The boundary layer depth is used to determine up to which height the released $CO_2$ may disperse following the vertical Gaussian plume model depending on, for example, distance to the source and atmospheric stability. It does not imply or assume that released $CO_2$ is instantly well mixed. See also our answer to R#17. However, as can be clearly seen from the in-situ vertical cross-section in the Fig. 5, the $CO_2$ increase is indeed reaching up to the highest available in-situ legs. Please keep in mind that the Fig. 4 on the other hand is conceptual and not based on actual data. Figure 4 was replace for the revised version to avoid confusion.

**R#23: P15L1 : Figure 11 shows that there was a significant decrease of the wind speed during the time of the in situ measurements. This should affect the intensity of the plume and I am surprised this was not discussed in the in-situ section.**

A: It is not completely clear what the reviewer means by "intensity of the plume". The in-situ method considers concentration and wind speed measured simultaneously, so decreased concentrations with higher wind speed will still yield the same flux.

**R#24: P17L4. Are there really any significant difference for the modelled wind speed over the 10 km area?**

A: The standard deviation over the measurement area for the model layer shown in Figure 8 is about 5.8%. This is mentioned in the revised version.

**R#25: P17L8. Section 5.2.4 is supposed to be a "Result" section. Yet, a large fraction of it describes the method**

A: Agreed. Was moved to Section 4.

**R#26: P17L16: It is said that the measurements are**

A: Unfortunately the reviewer's comment is not complete here. We have checked the corresponding part of the manuscript and did not identify any obvious issues.

**R#27: P18L4. Description of the wind speed estimate. It is said that a Gaussian profile for the concentration is assumed. Yet, the in situ measurement do not show such Gaussian profile. It would be nice to compare the assumed vertical distribution of CO2 with the in situ measurements.**

A: We repeat here, what we answered to comment R#2:
The Gaussian shape of the vertical distribution can only be observed close to the source. Due to the "reflection" of the plume off the surface and off the top of the boundary layer, the $CO_2$ gets mixed rather rapidly. About 2 km downwind of the source the $CO_2$ is well mixed according to the Gaussian model, i.e. on average and not necessarily for any snapshot in time. The in-situ measurements, for example, downwind of power plant Niederaußem (Fig. 5) are more than 2 km downwind of the source so that no distinct Gaussian shape in the vertical concentration profile is to be expected.

**R#28: Also, the fact that there is little vertical gradient in the wind speed makes this discussion somewhat unnecessary.**

A: That is to some degree true. On the other hand, a complete description of the method should involve the estimation of the average wind speed. For the revised version we condensed the discussion.

**R#29: How is done the weighting to derive a mean wind speed ?**

A: We extended the main text:
The emitted $CO_2$ was then distributed using a vertical Gaussian dispersion with the stability parameter resulting from the 2D horizontal Gaussian plume inversion model. This

information could be used to obtain an altitude weighted mean wind speed for the remote sensing cross sections through the plume based on relative concentrations per altitude layer.

**R#30: P18L18. "Very unstable atmospheric conditions". Is that consistent with the observed meteorological conditions on that day?**

A: The convective dispersion leads to unstable atmospheric conditions. Note also that the derived stability is an effective parameter that also subsumes other effects such as increased flew gas temperature or even changes in wind direction that may lead to additional plume broadening and dispersion. We are more explicit about that in the revised version.

**R#31: P23L22: The authors state the error analysis leads to an uncertainty on the order of 10% for the mass balance approach. This is in contradiction, I believe, with the results shown in Figure 15 that show larger variations for the various leg estimates. For instance, three legs can be used to estimate the emissions from Niederaussem. There is a factor of 2 between the largest and the smallest. This appears contradictory with the error analysis, in particular since**
**several of the error sources are biases and cannot explain a difference between the estimates from two nearby legs. I am surprised this is never discusses in the text.**

A: The reviewer is right. Our sensitivity study for the remote sensing mass balance approach did indeed not take into account any statistical errors. The magnitude of the flight track to flight track variability shows furthermore how critical it is to have a sufficient number of flight tracks to obtain an accurate estimate. As there are only few flight tracks per power plant, the error is naturally quite large. This is now discussed in the revised version, the error analysis was updated and the updated Figure 15 includes these uncertainties.

**R#32: P21L1: The whole section 5.2.5 is poorly written.**

A: The section was shortened and improved.

**R#33: P24L4: "can differ more than 20% for individual power plants". So what are you saying here ? Are you suggesting that the reported emissions shown in the paper (Figure 18) can be off by that much ?**

A: Not at all. We explicitly wrote:
"The error on power generation itself is generally about 1% (compare also Krings et al., 2011) and the annual error of derived emissions is required to be within 2.5% (European Commission, 2007). **The error for the time of the overflight is most likely not much larger**, although comparisons between U.S. inventories based on monitoring of stack gases with inventories based on emission factors can differ more than 20% for individual power plants (Ackerman and Sundquist, 2008)."

In summary:
  (1) We have no indication to believe that the error is larger than what is required.
  (2) There is a publication that found differences of 20% between different methods.

**R#34: In the following, I make comments on the figures. I strongly believe that several of them are not useful whereas other could bring additional information**
**Figure 1 : Figure 2 : Limited usefulness**

A: Figure 1 was updated to contain also the in-situ tracks and the new Figures 2 to 8 are replacing those that were questioned.

**R#35: Figure 4: Provide colour scale**

A: The old figures 4 and 5 are replaced by the new figures 2 to 8, which should be much clearer now. All color scales are provided.

**R#36: Figure 5 : Is this figure supposed to show the same data as in Figure 4 ?**

A: See above. No, it was not the same data, and we agree that the old Figure 5 was confusing because the concentrations between the cells are smoothed by the graphics program. We are sure that the new figures are much clearer and more consistent because it is clearly visible now which were the original measurements (Figures 2 & 3), and how they were treated on the grids. This allowed us to omit the old Fig. 4 which was only showing the concept.

**R#37: I cannot recognize any feature. I strongly recommend to show the value of the measurements within the circles that are used for the interpolations.**

A: See answers to R#35 and R#36.

**R#38: What is the link between this figure and the "wall" approach shown in Figure 4 ?**

A: See answers to R#35 and R#36.

**R#39: Figure 6: Definitely not useful. Not clear what is really shown (ie RMS of what, relative to what ?)**

A: This Figure justifies the 0.9% filter on the RMS of (model-measurement)/model. It is quite instructive documenting the good data quality. The Figure was removed anyhow. See also comment to R#19.

**R#40: Figure 7: Definitely not useful**

A: Was removed.

**R#41: Figure 8: Is it really XCH4 as indicated in the legend, or XCO2 ? Why no color scale ?**

A: Typo was corrected and the color scale added.

**R#42: Figure 9: Marginaly usefull. The text could simply say that the in situ measurements (potential temperature and aerosol) provide no useful information to determine the top of the boundary layer up to 1100 m**

A: The Figure was removed and the text updated accordingly.

**R#43: Figure 10 : Should definitely be presented in the "in situ" section, together with Figure 5, and not in the remote sensing section.**

A: Agreed. However, the Figure was removed for the revised version.

**R#44: Figure 11: Difficult to read. Values for the X-axis could be simpler (e.g. 5/10/15)**

A: Scale has been adjusted.

**R#45: Figure 12: Should be shown early on in manuscript.**

A: We agree. The Figure has been replaced by similar Figures shown at the beginning.

**R#46: Why is the color scale not adjusted to the data (no observation before 12)**

A: This is to have the same color scale for all associated plots making comparisons between plots easier.

**R#47: Figure 13: I suggest to reduce the range of the color bar to 0.99-1.02 and have color lines (Gaussian plume model) for 1.005, 1.01, 1.015 and 1.02**

A: Adding more lines will make the plot unreadable. In combination with the complementary Figure 14 which now contains also the model result (see below), the information from Fig. 13 should be sufficiently detailed.

**R#48: Figure 14: I strongly suggest to add, on each of these graphs, a line showing the result of the modeling according to the Gaussian plume approach. Also, add a horizontal line to show the 0.**

A: The Figure has been updated accordingly.

**R#49: Figure 15 : State explicitly in the legend that each symbol corresponds to a flux estimate derived from a given aircraft leg.**

A: Done.

**R#50: Figure 16 : Not useful**
**R#51: Figure 17 : Not useful**

A: For our analysis we make choices for both wind direction and grid size and considered it reasonable to justify our decision and investigate the impact and sensitivity. Fig. 16 is furthermore important because it shows that wind direction can in principal be fitted directly to the data, which we now explicitly point out in the revised version. We kept Figure 16 but removed Figure 17 as suggested.

**References:**

Ackerman, K. V. and Sundquist, E. T.: Comparison of Two U.S. Power-Plant Carbon Dioxide Emissions Data Sets, Environ. Sci. Technol., 42, 5688–5693, doi:10.1021/es800221q, 2008.

Gordon, M., Li, S.-M., Staebler, R., Darlington, A., Hayden, K., O'Brien, J., and Wolde, M.: Determining air pollutant emission rates based on mass balance using airborne measurement data over the Alberta oil sands operations, Atmospheric Measurement Techniques, 8, 3745–3765, doi:10.5194/amt-8-3745-2015, https://www.atmos-meas-tech.net/8/3745/2015/, 2015.

Krings, T., Gerilowski, K., Buchwitz, M., Reuter, M., Tretner, A., Erzinger, J., Heinze, D., Pflüger, U., Burrows, J. P., and Bovensmann, H.: MAMAP – A new spectrometer system for column-averaged methane and carbon dioxide observations from aircraft: retrieval algorithm and first inversions for point source emission rates, Atmos. Meas. Tech., 4, 1735–1758, doi:10.5194/amt-4-1735-2011, 2011.

Krings, T., Gerilowski, K., Buchwitz,M., Hartmann, J., Sachs, T., Erzinger, J., Burrows, J. P., and Bovensmann, H.: Quantification of methane emission rates from coal mine ventilation shafts using airborne remote sensing data, Atmos. Meas. Tech., 6, 151–166, doi:10.5194/amt-6-151-2013, 2013.

Young, S. J.: Detection and quantification of gases in industrial-stack plumes using thermal infrared hyperspectral imaging. Aerospace Report No. ATR-2002(8407)-1. El Segundo, Calif: The Aerospace Corporation, 2002.

---

## Author Comment (AC2) · 30 Sep 2017

**Comment by L. Golston**

We would like to thank L. Golston for the additional comments. In the following we carefully address the comments point by point. First we repeat the comment (C#...) and then give our response (A).

**C#1: The authors describe two methods for quantifying point source emissions, one based on in-situ mass balancing and one on remote sensing coupled with an inverse Gaussian plume model, along with field data comparing the two methods. I would like to add three comments on the description of the in-situ method, which I hope will help strengthen that section.**
**It was surprising that Gordon et al. 2015 (AMT 8:3745-3765) and Cambaliza et al. 2014 (ACP 14:9029-9050) are not discussed or even cited, given that they both investigate in detail the uncertainties of the in-situ aircraft mass balance methodology.**
**Gordon et al. 2015 discusses issues of interpolation, extrapolation, turbulent fluxes, and issues related to determining the background concentration in the context of determining emissions from an elevated source as was done here, while Cambalizia et al. 2014 also considers interpolation, boundary layer entrainment, and other effects. On page 4 transparency of interpolations and extrapolations is claimed as a benefit here, but seem less well developed than in either of those papers. The description of both on Page 8, is actually specifically not transparent and neither seems to be included in the error budget of Table 4. Part of the reason why the in-situ method is discussed in such detail seems to be because a variation on the mass balance method is presented, however the differences and benefits are not clearly distinguished in Section 4.1 or the results.**

A: Two papers where Cambaliza was a co-author were referenced. However, the additional references from Cambaliza et al. (2014), and Gordon et al. (2015) were studied now in detail and were helpful. In the revised manuscript, we discuss especially the link to points that Gordon et al. have mentioned.

**C#2: I also recalled that Figs 3, 4, and part of 2 are identical to Figure 3 in Hacker et al. 2016 (Animal Production Science 56:190-203), who cited the report from the authors of the current paper, Bovensmann et al. (2014). Since the figures are now also in Hacker et al., I think that the original Bovensmann et al. (2014) should be referenced here to avoid confusion.**

A: The new Figures 2 and 3 for the measurements and 4 to 8 for the gridding are much clearer now.

**C#3: Finally, it would be helpful to know whether the turbulent (5 Hz) could be resolved as indicated in the caption for Figure 3 or if there was attenuation, and how the inclusion of the turbulent flux compares to not including it.**

A: This is now explicitly done by providing our standard fluxes (averages plus inter- and extrapolations of local mass x wind) plus 'flux 2', which was calculated after the averaging of the mass- and wind-field by our method, and by Kriging.
Since a complete revision was performed, and the separation of more individual sources was possible, the results as displayed in table 3 and Figure 18 were updated. The details are presented in a separate supplement.

---

## Author Comment (AC3) · 30 Sep 2017

**Reviewer 2**

We would like to thank the reviewer for the helpful and detailed comments. In the following we carefully address the comments point by point. First we repeat the reviewer's comment (R#...) and then give our response (A).

**R#1:The manuscript "Airborne remote sensing and in-situ measurements of atmospheric CO2 to quantify point source emissions" by Krings et al presents results from an airborne campaign, inferring point source fluxes of CO2 using both mass balance approaches as well as a Gaussian plume modeling of remotely sensed total column averages.**
**Even though the data presented here is indeed interesting, I tend to agree with reviewer 1 that it often reads much like a report and would need restructuring and more concise (and precise!) language.**

A: We improved on the overall structure to meet the guidelines given by the two reviewers.

**R#2: At times, the authors get caught up in details that are not entirely relevant to the study at hand, e.g. Figs 9-12 are too detailed or not necessary (9-10) or misplaced in the respective section**

A: We removed Figures 6, 7, 9, 10, 17 and moved Figure 12 to the description of the target area, respectively. Figure 11, however, was left in the remote sensing section. Although it shows in-situ data, the plot specifically addresses the remote sensing analysis.

**R#3: I would suggest putting a general description of the domain as well as the data right at the beginning (e.g. showing MAMAP footprints as well as in-situ ground projections on the map in Figure 1. It would greatly help setting the stage for the discussion.**

A: Figure 1 has been updated accordingly.

**R#4: Some more specific comments: Abstract last sentence: this is a sudden topic break and needs some rephrasing**

A: We rephrased the abstract:
*Reliable techniques to infer greenhouse gas emission rates from localised sources require accurate measurement and inversion approaches. In this study airborne remote sensing observations of $CO_2$ by the MAMAP instrument and airborne in-situ measurements are used to infer emission estimates of carbon dioxide released from a cluster of coal fired power plants. The study area is complex due to sources being located in close proximity and overlapping associated carbon dioxide plumes. For the analysis of in-situ data, a mass balance approach is described and applied. Whereas for the remote sensing observations an inverse Gaussian plume model is used in addition to a mass balance technique. A comparison between methods shows that results for all methods agree within 10% or better for cases where in-situ measurements were made for the complete vertical plume extent. The computed emissions for individual power plants are in agreement with results derived from emission factors and energy production data for the time of the overflight.*

**R#5: Page 3, line 30: wheras (there are many small things like this or "straight forward", which is one word. I won't go into more details, the copy-editor should catch those at a later stage but some sentences are too literal translations from german.**

A: We carefully went through the manuscript to improve wording.

**R#6: Section 4: I think this is poorly described and justified and I urge the authors to consider revisiting the differences between their approach and the paper Levi Golston mentioned.**

A: This was done very extensively.

**R#7: E.g.**
**I) you mention "Kriging" is not necessarily the best suitable approach. If you provide critic, you have to back it up either with an analysis or a citation. There is also no real explanation what kind of interpolation schemes you are using (apart from the boundary voxels).**

**A:** As a main part of the revision, we did the whole calculations with our linear inter- and extrapolation method with only four rules, and with Kriging. Our method is now clearly described, and displayed in the new figures 4 to 8. There are two aspects to distinguish: (i) about the inter- and extrapolation. By using Kriging as another method we have shown that the difference is small. (ii) More important seems to average and interpolate fluxes, instead of averaging mass- and wind-fields before calculating the fluxes. Especially when the latter was done by Kriging, the deviation from the ensemble of other solutions is increasing, most likely due to the fact, that small artefacts in the individual fields are increasing the errors. Bottom line: The presented method is not better than Kriging, but, Kriging should not be regarded as the only option. This is also true after studying Gordon et al. (2015) in detail. The new text is much clearer in showing and discussing these details.
Since a complete revision was performed, and the separation of more individual sources was possible, the results as displayed in table 3 and Figure 18 were updated. The details are presented in a separate supplement.

**R#8: II) You mention that you include turbulent fluxes. This is very interesting and I was excited but then I didn't see any further analysis. Did you compute the differences with or without turbulent components? What is the relative error in your case? Can you plot c-mean(c) vs v-mean(v) for some voxels to show the correlations as expected for turbulent fluxes?**

A: Yes, this is done now. The different results can be found in a detailed Table as a supplement since it would be too much for Table 3 in the publication. The differences with and without turbulent fluxes were very small, and – surprisingly – negative, i.e. the turbulent contributions are rather from dilution (entrainment) than adding to the flux.

**R#9: III) Wind speed seems to be a dominant error, do the others actually matter? You will need to provide realistic estimates regarding kriging and turbulent fluxes, otherwise the reader won't be able to judge the importance (even though this specific case study might not lend itself to extrapolation to a general case). Page 5, line 9: As above, please show what error you incur by doing mean(v)*mean(c) vs. mean(v*c)**

A: This is now explicitly done by providing our standard fluxes (averages plus inter- and extrapolations of local mass x wind) plus 'flux 2', which was calculated after the averaging of the mass- and wind-field by our method, and by Kriging.

**R#10: Figure 4: Please add color-bar and make this a realistic example based on real data.**

A: The confusion about the old Figures 4 and 5 should now be eliminated. The new figures are much clearer and more consistent because it is clearly visible now which were the original measurements (Figs. 2 and 3), and how they were treated on the grids. This allowed to omit Fig. 4 which only showed the concept.

**R#11: How many data points to average do you typically have per voxel?**

A: A grid cell which was crossed once or twice is averaging the 5 Hz data (concentrations and wind), resulting in typically 10 to 20 data points when hres and zres were 100 m. However, the size of the cells was varied between 100 and 200 m for the sensitivity analysis, resulting in 10 to 40 points.

**R#12: Page 9, line 8: Conditio sine qua non: Even though I have a "Grosses Latinum", I had to look it up again. Please rephrase in plain english, esp. as it is here used in a rather trivial way, not warranting the grandiose latin phrase ;-).**

A: Was revised.

**R#13: One might argue though that precision "could" be important if it is really bad while accuracy won't matter. This could be a factor when flying very cheap instruments on small unmanned aerial vehicles near the plume. So I would keep the discussions as general as possible.**

A: We agree in principle. However, the absolute accuracy of the concentrations remains less important when subtracting the background based on the same measurements. This is also true for expensive and relatively stable instruments.

**R#14: Page 9, line 17: Chimneys: It would be good to discuss how well you could measure the fluxes if the emissions were to happen at the surface. What would this imply for the in-situ based approach and potential flight-paths.**

A: We discussed this in the referenced ESA report for $CH_4$, which was emitted from coal shafts close to the surface. Of course in such cases it is more important to have the lowest flight track as low as possible (50 m above ground with a special permission). However, in this campaign, where we flew to many sources for which we had no low-flying permission, this was restricted to 150 m. This was irrelevant for the $CO_2$ from high chimneys though. Another factor is the stability of the boundary layer. Since we have chosen daytimes for flying, when the convection reached at least the 150 mAGL, extrapolations with constant flux for the $CH_4$ (not in this paper), and diminishing or constant concentrations for the $CO_2$ (both options for checking the sensitivity) were applied. The constant flux for the $CH_4$ was applied because an expected enhancement of concentrations near the surface was compensated by the diminishing wind speed. For more reliable results for $CH_4$ we would prefer mobile measurements near the surface, which some of the authors did during a separate campaign in summer 2016. This aspect is now mentioned in the revised text.

**R#15: Figure 5: This figure confuses me. I "assume" the dots are actual measurement locations.**

A: See **R#10**.

**R#16: Given what you wrote, there is a constant extrapolation to the surface. However, it doesn't look that way for the second little intrusion at x=0. Also, there are a couple of local maxima in between dots. You need to explain the interpolation scheme and this would a good place to compare against kriging or other interpolation schemes. Also, if the dots are measurements, please color-code them with the actual measurement values at that x-y position. This will help evaluate the interpolated fields better. A last point: Why is this continuous on x and y? Wouldn't it make sense to sketch out the actual grid boxes here as well?**

A: Fully done. Figure 5 was confusing and misleading by several reasons: (i) Showing concentrations instead of the locally measured fluxes is potentially misleading; (ii) the old figure added interpolation & smoothing from the graphics package. The new style of cross sections in Figures 4 through 8 is much clearer.

**R#17: Page 12, line 3: Please add citation for proxy method (this is not common knowledge).**

A: We added the references to the revised manuscript:
*See Frankenberg et al. (2005) and Schepers et al. (2012) for more information on the proxy method and Krings et al. (2011) for its application on MAMAP measurements.*

**R#18: Figure 6: I think the figure itself doesn't tell more than the text, could be skipped.**

A: Figure 6 has been removed.

**R#19: Same with Fig. 7**

A: Figure 7 has been removed.

**R#20: Page 14, line 17: So in essence, you don't really need the wind speeds in this case as the error is rather small?! Ideally, you won't always need both aircraft. If there is confidence in modeled winds, it would be a good sign for future remote-sensing only campaigns.**

A: We agree with the reviewer. Not having to use additionally measured in-situ wind speed would indeed be a huge step forward. A wider and systematic analysis on the accuracy of the model wind would indeed be very interesting but can of course not be accomplished within the present study. We added a short discussion about this in the manuscript:

*The analysis in this study shows an overestimated model wind speed of about 6% (or about 0.4 m/s) which is smaller than the uncertainty on wind speed. So in this case relying on the model alone may be sufficient. In a former study of similar setup (Krings et al., 2013) the error was about 10% or (0.7 m/s). A wider and systematic analysis on the accuracy of the model wind is needed to assess to what extent additional wind (profile) measurements are*

*dispensable. This will also become more relevant with regard to observations of localized sources by current and upcoming satellite missions with increased accuracy and spatial resolution. In these cases additional wind measurements will generally not be available.*

**R#21: Figure 10: Weird x-spacing (value 493?). also better to use same x-scale for both subplots.**

A: We removed the Figure completely as suggested in R#2.

**R#22: Page 21, line 10 +/-: Wouldn't you ideally fit a Gaussian model with a vertical windspeed profile? This would rather directly model the total column AND the wind-profile.**

A: Currently our model is set up to work with an average wind speed. The direct utilisation of the vertical wind profile u(z) for the inversion would be an interesting experiment. This would basically mean fitting the measurements to the sum of a vertically piecewise (or even continuously) changing Gaussian plume model (ignoring second order lateral and temporal wind speed variations in a first step). However, as Reviewer 1 pointed out that the discussion is "somewhat unnecessary", since in this case the vertical gradient in wind speed is not very strong we do not want to extend the discussion on this here and leave that to future work on the topic.

**R#23: How high is your Gaussian profile extending to the vertical? That might be a plot to add (or is it just 2D in x and y?).**

A: The Gaussian model to determine the average wind speed is 2D in along wind and vertical directions (x and z). With the same reason as above we do not want to add an additional plot for this but briefly discuss the vertical distribution as a function of downwind distance:
*At the first remote sensing leg 700m downwind of power plant Niederaußem, the plume reaches about 1 km height, and at 2 km downwind distance the CO2 is already well mixed according to the plume model which represents a temporal average.*

**R#24: Page 23, line 19: "This is because they to a good extent cancel out..." "largely" cancel out?**

A: Done.

**R#25: Page 26, line 20: I think they gase don't need to be inert, just have lifetimes much longer than the time between emission and measurement. I would guess even NOx emissions could work with an "inert" assumptions on this very local scale.**

A: Agreed. Has been clarified in the text of the revised version.

**R#26: As a last general point: Please try to re-structure somewhat to bring out the key messages in a more concise way (and illustrate better how your in-situ inversions differ from others).**

A: We agree and improved on the description as indicated in our answers above and to Reviewer 1.

**R#27: At the end, provide a more generic overview of both flux estimates and**

**its pro/cons and path forward. This could extend to a discussion using high-resolution mapping like the cited AVIRIS-NG papers (which should be cited at page 26, line 9). Last but not least my sincere apologies for the late review.**

A: Done. The discussion will be along the lines of what we also wrote as answer to Reviewer 1:

[revised manuscript text omitted]

---

## Author Response (AR2)

We would like to thank the referee for the helpful comments. In the following we address the comments point by point. First we repeat the reviewer's comment (R#...) and then give our response (A). We also provide a markup version of the revised manuscript.

**R#1: This paper is a large revision from the original submission. It is indeed largely improved both in terms of clarity and presentation. The authors have adequately answered to the reviewers comments. The paper can then be published with little corrections**

**My only significant comment concerns the plum intermittency. This intermittency is very clearly observed in the in-situ measurements and also on the remote sensing data. This intermittency is probably the largest contributor to the total uncertainty for a the emission estimate from a single track. Arguably, it is a random uncertainty that decreases with the number of samples. As the primary contributor to the uncertainty, there should be more efforts in its quantification.**

A: The uncertainty caused by the plume intermittency is indeed a major error source. We quantified the uncertainty on the remote sensing inversion result for each individual power plant in Figure 15. For in-situ we acknowledged the intermittency of the plume as error source in the manuscript (P12L25 ) but did not separately quantify it as a consequence of the approach taken.

**R#2: Abstract : The abstract should include the error estimates, not only the observed differences with "truth"**

A: Done:

*A comparison between methods shows that results for all methods agree within 10% or better with uncertainties of 10% to 30% for cases where in-situ measurements were made for the complete vertical plume extent.*

**R# 3: P10L2 « 200ppm and more ». Figure 2 shows that the plume concentration increase are, in most cases, much less than 200 ppm.**

A: We rephrased to a slightly more conservative statement which however does not change our conclusions:
*The background concentration for this case study was relatively easy to find, and the results are not sensitive to it because the peak plume concentrations were high above the background (100 ppm and more; see Fig. 2).*

We also adapted the caption of Figure 3 in that sense:

*The background concentration was determined to be at about 392 to 393 ppm, which is a small uncertainty compared to the maximum plume concentrations of about +100 ppm).*

**R#4: P10L6 « lower background concentration ». Why should one expect lower background concentration towards elevated layers ?**
A: We slightly rephrased the statement. Please note that the important point here is that the background concentration at emission height is most relevant and not the background at plume height.
*...resulting typically in slightly decreasing background concentrations with altitude in this region: The emissions were injected at low altitudes into the background concentrations that were present there. When such a plume is rising into lower (or higher) background concentrations, taking the local background would lead to an over- or underestimation of the emission.*

**R#5: Figure 13 : The Gaussian plume model is valid everywhere, not only on locations where there is a valid measurement. Thus the red line on the figure should be continuous and not with a linear interpolation between data points. Conversely, there is no reason no link the datapoints with a line.**

A: Done.

**R#6: P25L8-10. I could not understand the statement about American power plants as a counter justification for the stability of the PP emission**

A: We rephrased:

*The error for the time of the overflight is most likely not much larger. However, a study by Ackerman and Sundquist (2008) shows for individual U.S. power plants that inventories based on monitoring of stack gases and inventories based on emission factors can in principal differ more than 20%.*

**R#7: P26L4. This sentence is linked to the general statement about the intermittency and its impact on the uncertainty. It should attempt at providing quantitative values**

A: Instead of repeating the quantitative numbers here we referenced Fig. 15 which explicitly states the requested uncertainties. Please see the original paragraph from the manuscript:

[revised manuscript text omitted]